# TOWARDS MONOTONIC IMPROVEMENT IN IN-CONTEXT REINFORCEMENT LEARNING

## ABSTRACT

In-Context Reinforcement Learning (ICRL) has emerged as a promising paradigm for developing agents that can rapidly adapt to new tasks by leveraging past experiences as context, without updating their parameters. Recent approaches train large sequence models on monotonic policy improvement data from online RL, aiming to achieve continued improvement in testing time performance. However, our experimental analysis reveals a critical flaw: these models cannot demonstrate continued improvement like the training data during testing time. Theoretically, we identify this phenomenon as *Contextual Ambiguity*, where the model's own stochastic actions can generate an interaction history that misleadingly resembles that of a sub-optimal policy from the training data, initiating a vicious cycle of poor action selection. To resolve the Contextual Ambiguity, we introduce *Context Value* into training phase and propose **Context Value Informed ICRL** (CV-ICRL). CV-ICRL uses Context Value as an explicit signal representing the ideal performance theoretically achievable by a policy given the current context. As the context expands, Context Value could include more task-relevant information, and therefore the ideal performance should be non-decreasing. We prove that the Context Value tightens the lower bound on the performance gap relative to an ideal, monotonically improving policy. We further propose two methods for estimating Context Value at both training and testing time. Experiments conducted on the Dark Room and Minigrid testbeds demonstrate that CV-ICRL effectively mitigates performance degradation and improves overall ICRL abilities across various tasks and environments. The source code and data of this paper are available at `https://anonymous.4open.science/r/towards_monotonic_improvement-E72F`.

## 1 INTRODUCTION

As reinforcement learning (RL) algorithms are increasingly deployed in diverse and dynamic environments, there is a growing demand for methods that can generalize across tasks and adapt efficiently to novel situations, a challenge that current RL algorithms still struggle to address (Finn et al., 2017; Cobbe et al., 2019). A promising direction toward this goal is In-Context Reinforcement Learning (ICRL), where agents adapt to unseen tasks purely through interaction with the environment without updating model parameters, only by leveraging past experiences provided as context (Brown et al., 2020; Chan et al., 2022). Current advanced ICRL methods leverage offline datasets containing trajectories of increasing policy quality, which aim to a continue performance improvement in testing time. For instance, Algorithm Distillation (AD) and its subsequent works, including our approach, are trained using continuously enhanced trajectories, generated from online RL algorithms and demonstrations (Laskin et al.; Huang et al.).

However, a critical gap emerges between this idealized continuously improved training data and test-time performance. Through case study, we find that these ICRL methods suffer from severe performance regression during inference, which cannot achieve monotonic improvement as training dataset shows (Figures 1 and 2). Theoretically, we further analyze this phenomenon and find that the single action sampling for each context at testing time can violate the implicit assumption of sufficient sampling needed to average out stochasticity. Such a violation might lead the model to misidentify its own skill level, initiating a vicious cycle of performance degradation. We name this violation as **Contextual Ambiguity** problem of the ICRL methods, which means a single stochas-

tically poor action can generate a context that misleadingly resembles a history from a weaker, sub-optimal policy.

To fundamentally address the Contextual Ambiguity problem, we introduce a theoretical construct, the **Context Value** ($V_C$) and propose **Context Value Informed ICRL** (CV-ICRL). We define the context value as the ideal performance theoretically achievable by a policy given the information in context $C$, i.e., $V_C = J(\pi_C^*)$. The core purpose of introducing the Context Value is to provide an unambiguous quality label for the current context $C$. This label allows the policy to bypass the perilous inference from noisy historical interactions and instead adjust its behavior based on this value signal. We theoretically prove that the introduction of Context Value mitigates the degradation caused by ambiguity and tightens the performance bound between the learned policy and the context-optimal policy, thereby providing stronger guarantees of performance monotonicity during inference.

In practice, CV-ICRL estimates the context-optimal policy $\pi_{C_i}^*$ with the context generated by the behavior policy $\pi_i$, and consequently, estimate the Context Value $V_{C_i}$ as its expected return. Building on this, we propose two methods for estimating the Context Value at both training and testing time. Moreover, we prove that when the estimation errors of $\pi_{C_i}^*$ and $V_{C_i}$ are sufficiently small, the tightened performance bound still holds. We conduct experiments on tasks in the Darkroom and Minigrid environments. The results demonstrate that our proposed method successfully addresses performance degradation, improves the stability of test-time performance improvement, and achieves significant gains in metrics such as average episode return.

In summary, our main contributions are given below:

- We identify that AD-like ICRL methods often suffer from performance degradation at testing time, failing to preserve the monotonic improvement property of training data. We analyze this phenomenon and attribute it to context ambiguity, where randomness in test-time sampling misleads decision-making.

- We introduce Context Value as a measure of context quality and propose CV-ICRL. We provide a theoretical guarantee that incorporating it yields a tighter performance bound between the learned policy and the context-optimal policy, thereby better preserving monotonicity.

- Experiments demonstrate that CV-ICRL effectively mitigates performance degradation and improves overall performance. Moreover, our study is the first to show that AD-like ICRL methods exhibit strong generalization across different task types in Minigrid.

## 2  RELATED WORKS

In-context reinforcement learning (ICRL) is a subfield of meta-RL that operates in few-shot, multi-task settings (Beck et al., 2023), where an agent adapts by conditioning on recent trajectory context without gradient updates. In practice, ICRL is often instantiated with causal Transformers that model long-horizon context and act autoregressively. This paradigm was popularized by Decision Transformer (DT) (Chen et al., 2021), which frames RL as conditional sequence modeling—predicting the next action from a history of states, actions, and rewards via supervised learning on trajectory data—in lieu of value iteration or policy gradients. Building upon this foundation, Prompt-DT (Xu et al., 2022) showed that by injecting contextual prompts, such as natural language instructions or goal specifications, into the input sequence, a single pretrained model could be guided to solve various tasks without fine-tuning. This use of contextual information to steer behavior which represents an early form of the ICRL method.

Algorithm Distillation (AD) (Laskin et al.) was the first approach to leverage a causal Transformer to address the problem. The core idea of AD is to distill the online RL learning process into a large causal model via supervised learning. Since then, a series of AD-like methods have been proposed, all of which share the same training paradigm, that the ICRL model is trained on continuously enhanced trajectories. $AD^\epsilon$ (Zisman et al.) demonstrates that actual trajectories from online RL are not strictly required; instead, training trajectories can be simulated by sampling from a noised model and gradually reducing the noise level. Agentic Transformer (AT) (Liu & Abbeel, 2023) organizes training trajectories by their episode rewards, aligning them with a chain of hindsight targets. Building on

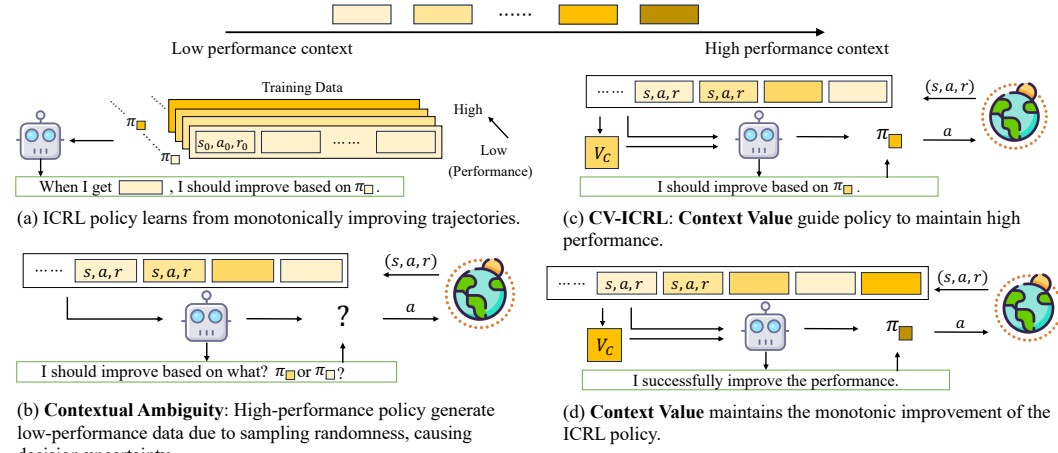

(a) ICRL policy learns from monotonically improving trajectories.

(b) **Contextual Ambiguity**: High-performance policy generate low-performance data due to sampling randomness, causing decision uncertainty.

(c) **CV-ICRL**: **Context Value** guide policy to maintain high performance.

(d) **Context Value** maintains the monotonic improvement of the ICRL policy.

Figure 1: ICRL policy is trained on monotonically improving trajectories (a). However, **Contextual Ambiguity** breaks monotonic improvement at testing time (b). We propose **CV-ICRL** (c), in which **Context Value** helps the ICRL policy to preserve monotonic improvement (d).

AT, In-context Decision Transformer (IDT) (Huang et al.) highlights the computational challenges of processing long-horizon inputs in Transformer models, and introduces a hierarchical decision-making structure to model longer contexts. Our method follows AD-like training frameworks, but differs in that we explicitly addresses the gap between training data and test-time performance: we identify that Contextual Ambiguity at testing time can lead to performance degradation, and propose an improved algorithm to mitigate this issue.

Beyond the AD-like methods, several alternative approaches to ICRL have also been explored, which provide broader perspectives for advancing this field. Decision-Pretrained Transformer (DPT) (Lee et al., 2023) adopts a posterior-sampling perspective, using optimal actions as supervised signals and acting optimally for a task sampled from the posterior, but it requires access to task-optimal policies and struggles with out-of-distribution generalization. Scalable In-Context Q-Learning (SICQL) (Liu et al., 2025) takes an offline Q-learning approach, enabling explicit value estimation and credit assignment to extract high-quality actions even from suboptimal trajectories. Besides these supervised pretraining methods, there are also reinforcement pretraining methods that usually involves online environment interactions (Moeini et al., 2025). AMAGO (Grigsby et al., a;b) employing an actor-critic framework and off-policy learning design to train long-sequence transformers in a scalable and fully end-to-end manner.

## 3 BACKGROUNDS

### 3.1 MARKOV DECISION PROCESS

We model reinforcement learning (RL) as a Markov Decision Process (MDP) $\mathcal{M} = (\mathcal{S}, \mathcal{A}, T, R, \gamma)$, where $T(s'|s, a)$ is the transition probability function, $R(s, a)$ the reward, and $\gamma \in [0, 1)$ the discount. At time $t$, the agent observes $s_t$, samples $a_t \sim \pi(\cdot|s_t)$, receives $r_t = R(s_t, a_t)$, and transitions $s_{t+1} \sim T(\cdot|s_t, a_t)$. The objective in RL is to find an optimal policy $\pi^*$ that maximizes the expected sum of discounted rewards, denoted as $J(\pi)$. $J(\pi)$ is defined as the expected return starting from an initial state $s_0$ sampled from a distribution $p_0$, i.e. $J(\pi) = \mathbb{E}_{s_0 \sim p_0}[V^\pi(s_0)]$, where $V^\pi(s) = \mathbb{E}_{\pi, T}\left[\sum_{t=0}^\infty \gamma^t R(s_t, a_t) \mid s_0 = s\right]$, and the expectation $\mathbb{E}_{\pi, P}$ is over the trajectory distribution induced by the policy $\pi(a_t|s_t)$ and the environment's dynamics $P(s_{t+1}|s_t, a_t)$.

### 3.2 IN-CONTEXT REINFORCEMENT LEARNING

Instead of single, fixed MDP for RL, In-Context Reinforcement Learning (ICRL) considers a distribution of tasks $p(\tau)$, where each task $\tau$ is a distinct MDP $(\mathcal{S}, \mathcal{A}, T_\tau, R_\tau, \gamma)$. The core challenge is to train a single, general policy that can quickly infer the dynamics and reward structure of a new task from a small amount of interaction history and then act near-optimally. An ICRL policy, accordingly,

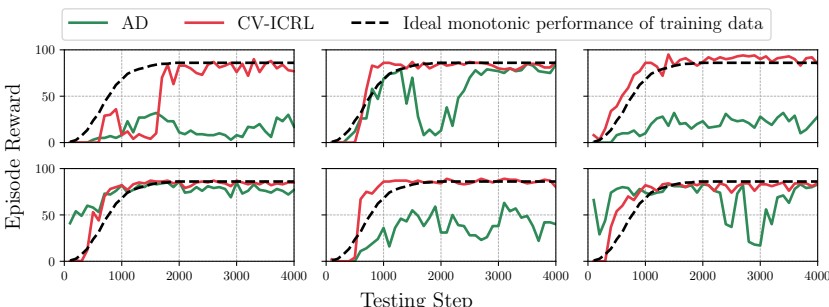

Figure 2: Across 6 different Dark Room tasks, AD struggles to maintain the ideal monotonic performance as training data shows, and CV-ICRL successfully maintains it.

is conditioned not just on the current state, but on the history of recent interactions. A history or context, $C_t$, is a sequence of state-action-reward tuples: $C_t = (s_0, a_0, r_0, \ldots, s_{t-1}, a_{t-1}, r_{t-1}, s_t)$. The ICRL policy $\pi^{\text{ICRL}}$ takes this context rather than single-step state to predict the next action: $\pi^{\text{ICRL}}_{C_t} = \pi^{\text{ICRL}}(a_t|C_t)$. For convenience, we labeled $\pi^{\text{ICRL}}_{C_t}$ as $\pi_{C_t}$ in the following text. The model must learn to recognize patterns within the trajectory context to deduce the underlying MDP dynamics and rewards, effectively performing "in-context" adaptation without updating its network weights. Similarly to $J(\pi)$, the objective of ICRL on task $\tau$ is $J(\pi^{\text{ICRL}}; \tau) = \mathbb{E}_{s_0 \sim \rho_{0,\tau}} \left[ V^{\pi^{\text{ICRL}}}(s_0; \tau) \right]$, where $V^{\pi^{\text{ICRL}}}(s; \tau) = \mathbb{E}_{\pi^{\text{ICRL}}, T_\tau} \left[ \sum_{t=0}^{\infty} \gamma^t R_\tau(s_t, a_t) \mid s_0 = s \right]$. Then $J(\pi^{\text{ICRL}}) = \mathbb{E}_{\tau \sim p(\tau)} \left[ J(\pi^{\text{ICRL}}; \tau) \right]$.

### 3.3 Contextual Ambiguity in In-Context RL

AD-like ICRL algorithms are typically trained under a set of assumptions that guide the learning process. These assumptions are formally described as follows:

**Assumption 1** (Properties of Training Data in ICRL)**.** Algorithm Distillation-like ICRL policies are trained on datasets generated from a sequence of monotonically improving expert policies, $\{\pi_0, \pi_1, \ldots, \pi_T\}$, with the following assumptions:

1. **Generative Process:** Each context $C_t$ is generated by a sequence of actions sampled from corresponding source policies, $a_t \sim \pi_t(\cdot|s_t)$.

2. **Sufficient Sampling:** The training set contains sufficient samples from each policy $\pi_i$ to allow the model to learn a robust mapping from the contexts $C_t$ to the corresponding target policy $\pi_t(\cdot|s_t)$.

3. **Monotonic Improvement:** The performance of the source policies, measured by a return function $J(\cdot)$, is monotonically non-decreasing, i.e., $J(\pi_i) \leq J(\pi_j)$ for all $i < j$.

Given these assumptions, one might expect that the model should also exhibit monotonic performance improvement during testing, similar to the behavior seen during training. However, this ideal scenario does not always hold in practice.

To investigate this, we conducted a case study in the Dark Room environment and observed that the AD model does not maintain monotonic improvement in episode return as the test timestep increases, as shown in Figure 2. In some cases, the model's performance even failed to recover to previously achieved levels. We attribute this discrepancy to **Contextual Ambiguity**, as illustrated in Figure 1(b). Due to sampling randomness, short-term contexts may contain low-reward samples that mislead the model and induce suboptimal decisions; consequently, the model misidentifies its stage and transitions prematurely to a more advanced policy, exacerbating performance degradation. In Appendix B, we provide additional insights into Contextual Ambiguity and elaborate on its resulting implications for performance.

# 4 TOWARDS MONOTONIC IMPROVEMENT IN IN-CONTEXT RL

## 4.1 CONTEXT VALUE: A STEP TOWARDS MONOTONIC IMPROVEMENT

Ideally, as the context expands, it should contain more task-relevant information ideally. An ideal ICRL policy would be able to infer increasingly useful information from the context, thereby producing a policy whose performance is non-decreasing. Here, we provide the definition of the context-optimal policy.

**Definition 1** (*Context-optimal policy*). For a given context $C$ sampled from task $\tau$, the *context-optimal policy* $\pi_C^*$ is the oracle policy that yields the highest expected return that an ICRL policy can achieve, only based on $C$, without any other information of $\tau$.

$$\pi_C^* = \arg\max_\pi \mathbb{E}_{\tau,C,R}[R(\tau, \pi(C)], \quad \text{s.t.} \quad I(\tau; \pi) \leq I(\tau; C), \tag{1}$$

where $I(\tau; C)$ denotes the mutual information between the task and the context, and $I(\tau; \pi) \leq I(\tau; C)$ ensures that the policy does not exploit any information beyond what is available in the context.

**Definition 2** (*Context Value*). The *Context Value*, denoted as $V_C$, represents the ideal performance theoretically achievable by a policy given the information in context $C$, i.e. $V_C = J(\pi_C^*)$.

**Property 1** (Monotonicity of $V_C$). Let $C'$ be a new context formed by adding a data sample $(s, a, r)$ from task $\tau$ to the original context $C$, i.e. $C' = C \cup \{(s, a, r)\}$. Because $C'$ contains more information about task $\tau$, we have:

$$J(\pi_{C'}^*) \geq J(\pi_C^*) \quad \text{and therefore} \quad V_{C'} \geq V_C. \tag{2}$$

**Why introduce the Context Value?** The Context Value is introduced to resolve the Contextual Ambiguity. It provides an unambiguous quality label for the current context $C$, enabling the policy to bypass the perilous inference from historical interactions, thereby breaking the cycle of performance degradation.

**Property 2** (Ideal performance monotonicity of ICRL policy via Context Value). Let $\pi_C(\cdot|C, V_C)$ be a policy conditioned on both the context and its oracle value. If we have access to the oracle Context Value $V_C = J(\pi_C^*)$, and the policy $\pi_C(\cdot|C, V_C)$ perfectly learns to output the actions of the context-optimal policy, i.e., $\pi_C(\cdot|C, V_C) = \pi_C^*(\cdot|C)$, then its expected return will be optimal: $J(\pi_C) = V_C = J(\pi_C^*)$. And because of the monotonicity of $V_C$, we have:

$$J(\pi_{C_j}) = V_{C_j} \geq V_{C_i} = J(\pi_{C_i}), \quad \forall j > i. \tag{3}$$

In above property, we made an idealized assumption, that we can access the context-optimal policies $\pi_C^*$ for each given $C$ and the ICRL model $\pi_C$ perfectly learns $\pi_C^*$ for both naive ICRL methods and our method. However, in real practice, the learning process itself induces a performance gap $E$ (or an expected return error) between the learned policy $\pi^{\text{ICRL}}$ and the monotonically improved optimal policy $\pi_C^*$, that is

$$E = |J(\pi_C^*) - J(\pi_C)|. \tag{4}$$

**Theorem 1** (**Improved Performance Bound**). Let the worst-case performance errors for the baseline policy and the value-informed policy be bounded as follows:

$$\sup_C |J(\pi_C^*) - J(\pi_C(\cdot|C))| \leq \frac{2r_{\max}}{(1-\gamma)^2} \epsilon_{\text{base}}, \tag{5}$$

$$\sup_C |J(\pi_C^*) - J(\pi_C(\cdot|C, V_C))| \leq \frac{2r_{\max}}{(1-\gamma)^2} \epsilon_V, \tag{6}$$

where $\epsilon_{\text{base}} = \sup_C D_{TV}(\pi_C^*(\cdot|C) \| \pi_C(\cdot|C))$, and $\epsilon_V = \sup_C D_{TV}(\pi_C^*(\cdot|C) \| \pi_C(\cdot|C, V_C))$

Then we can prove that $\epsilon_V < \epsilon_{\text{base}}$. Thus the **upper bound** for the value-informed policy $\pi_C(\cdot|C, V_C)$ is **strictly tighter** than the bound for the baseline policy $\pi_C(\cdot|C)$.

The proof is given in Appendix A.1.

## 4.2 Context Value Informed ICRL and Practical Algorithms

In contrast to the idealized formulation of ICRL where the oracle context value $V_{C_i}$ and context-optimal policies $\pi^*_{C_i}$ are assumed available, the practical setting only provides an offline dataset of contexts $C_i$ paired with their observed policies. This mismatch raises the key challenge: how to leverage such limited supervision to approximate the underlying context values and thereby improve adaptation. To bridge this gap, we introduce **Context Value Informed ICRL (CV-ICRL)**, a new procedure that augments standard ICRL by explicitly estimating the latent value of each context and incorporating it into testing time policy inference. The design follows from Theorem 1, leading to a corollary that shows performance can be provably improved when the estimation errors of both the context value $\widehat{V_C}$ and the context-optimal policy $\widehat{\pi^*_C}$ are sufficiently small. This reformulation establishes CV-ICRL as a principled extension of ICRL, equipped with a practical pathway for context value estimation and policy refinement under offline data constraints.

**Corollary 1** (Improved performance bound under estimated $\widehat{V_C}$). Let $\widehat{V_C}$ be the estimator of $V_C = J(\pi^*_C)$, and the worst-case performance errors be bounded as follows:

$$\sup_C |J(\pi^C) - J(\pi_C(\cdot|C, \widehat{V_C}))| \leq \frac{2r_{\max}}{(1-\gamma)^2}\epsilon_{\hat{V}}, \tag{7}$$

where $\epsilon_{\hat{V}} = \sup_C D_{TV}(\pi^*_C(\cdot|C)\|\pi_C(\cdot|C, \widehat{V_C}))$. Then $\epsilon_{\hat{V}} < \epsilon_{\text{base}}$ if the estimation errors of $\widehat{V_C}$ and $\widehat{\pi^*_C}$ are sufficiently small. The proof and required bounds are provided in Appendix A.2.

We propose two practical algorithms that share a common method for estimating $V_C$ at training time and differs at testing time. At training time, we estimate the context-optimal policy of context $C_i$ as the policy $\pi_i$ from the dataset, i.e. $\widehat{\pi^*_{C_i}} := \pi_i$. Consequently, our estimation for the Context Value $\widehat{V_{C_i}}$ is the expected return of this target policy: $\widehat{V_{C_i}} := J(\pi_i)$. At testing time, as we cannot access the expected return of the behavior policy $J(\pi_i)$, we propose two different ways to estimate it:

1. **CV-ICRL-$\phi(C)$** (Estimate $V_C$ through $C$): We parameterize $V_C$ as a function of context $\phi(C)$, implemented as an auxiliary output head in the Transformer-based ICRL model. During training, the source policy return $J(\pi_t)$ serves as the supervision signal for this head. This design enables estimated $V_C$ to adapt to task information embedded in the context, but the ambiguous contexts may lead to inaccurate estimates.

2. **CV-ICRL-$\phi(t)$** (Estimate $V_C$ through timestep): Motivated by the premise that Context Value should ideally increases as more task-relevant information is gathered over time, we tied estimated $V_C$ to the monotonically increasing variable, timestep. This guarantees monotonicity and robustness against context ambiguity, as $\phi(t)$ evolves independently of contexts. However, it may not adapt to task difficulty, leading to potential misalignment with the true Context Value.

Further details and pseudocodes are provided in Appendix C.

## 5 Experiments

### 5.1 Experimental Setup

**Environments.** We use Dark Room (Laskin et al.) and Minigrid (Chevalier-Boisvert et al., 2023) to evaluate our methods. Dark Room is a commonly used environment for ICRL algorithms, where Minigrid presents a more challenging and diverse benchmark than Dark Room. Minigrid tasks feature underlying MDPs that differ not only in their reward functions but also in their observation spaces. Furthermore, the consistent observation space across different task families in Minigrid allows us to rigorously test the model's cross-task generalization capabilities.

**Unseen Task and Unseen Task Types of each Environment.** For Dark Room that has only one task type, we test ICRL policies on 20 unseen tasks that did not appear in training datasets. For Minigrid, within each task type (e.g. LavaCrossingS9N3), we train baselines and CV-ICRL methods in 400 tasks and test on 20 unseen tasks. Additionally, we test 4 unseen task types to demonstrate the generalization capabilities of ICRL policies at the task-type level. More details of unseen tasks and task types are provided in Appendix D.1.

Table 1: We conduct a comprehensive set of experiments in 6 different types of tasks in the Minigrid. Details of these 6 tasks are provided in Appendix D.1. We report the mean and variance for 20 seeds (corresponding to different unseen tasks) of AER, LER, and Degradation Frequency for each task types. The Degra. Freq. for IDT is omitted as it fails in BlockedUnlockPickup. The best results are **in bold** and the second-best are underlined.

| Task Type | Metric | AD | AD-$\epsilon$ | IDT | CV-ICRL-$\phi(t)$ | CV-ICRL-$\phi(C)$ |
|---|---|---|---|---|---|---|
| LavaCrossing S9N3 | AER | $0.918 \pm 0.035$ | $0.902 \pm 0.059$ | $0.915 \pm 0.057$ | $\mathbf{0.934 \pm 0.027}$ | $0.921 \pm 0.051$ |
| | LER | $0.933 \pm 0.051$ | $0.936 \pm 0.052$ | $\underline{0.945 \pm 0.020}$ | $\mathbf{0.948 \pm 0.015}$ | $0.939 \pm 0.026$ |
| | Degra. Freq. (%) | $\underline{3.706 \pm 4.456}$ | $6.149 \pm 9.108$ | $5.590 \pm 8.810$ | $\mathbf{2.422 \pm 4.067}$ | $5.228 \pm 8.787$ |
| LavaCrossing S9N2 | AER | $0.937 \pm 0.027$ | $0.918 \pm 0.032$ | $0.898 \pm 0.097$ | $\mathbf{0.944 \pm 0.016}$ | $\underline{0.943 \pm 0.015}$ |
| | LER | $\mathbf{0.954 \pm 0.012}$ | $0.946 \pm 0.034$ | $0.927 \pm 0.060$ | $0.949 \pm 0.024$ | $\mathbf{0.954 \pm 0.011}$ |
| | Degra. Freq. (%) | $2.101 \pm 2.501$ | $3.524 \pm 3.735$ | $8.213 \pm 12.041$ | $\mathbf{0.815 \pm 0.872}$ | $\underline{1.557 \pm 1.289}$ |
| SimpleCrossing S9N3 | AER | $0.929 \pm 0.067$ | $0.152 \pm 0.119$ | $0.218 \pm 0.181$ | $\underline{0.942 \pm 0.016}$ | $\mathbf{0.945 \pm 0.015}$ |
| | LER | $0.941 \pm 0.029$ | $0.396 \pm 0.252$ | $0.529 \pm 0.226$ | $\mathbf{0.950 \pm 0.015}$ | $\underline{0.949 \pm 0.012}$ |
| | Degra. Freq. (%) | $3.428 \pm 8.615$ | $81.378 \pm 10.588$ | $70.875 \pm 15.810$ | $\mathbf{1.398 \pm 1.863}$ | $\underline{1.561 \pm 1.880}$ |
| SimpleCrossing S11N5 | AER | $0.865 \pm 0.124$ | $0.138 \pm 0.107$ | $0.323 \pm 0.296$ | $\underline{0.897 \pm 0.070}$ | $\mathbf{0.902 \pm 0.076}$ |
| | LER | $0.886 \pm 0.142$ | $0.329 \pm 0.242$ | $0.587 \pm 0.331$ | $\underline{0.905 \pm 0.082}$ | $\mathbf{0.921 \pm 0.066}$ |
| | Degra. Freq. (%) | $15.754 \pm 15.686$ | $73.744 \pm 12.312$ | $67.190 \pm 26.947$ | $\underline{13.694 \pm 13.729}$ | $\mathbf{13.209 \pm 13.108}$ |
| BlockedUnlock Pickup | AER | $0.911 \pm 0.160$ | $0.219 \pm 0.370$ | $0.000 \pm 0.000$ | $\underline{0.952 \pm 0.008}$ | $\mathbf{0.955 \pm 0.010}$ |
| | LER | $0.911 \pm 0.209$ | $0.057 \pm 0.134$ | $0.000 \pm 0.000$ | $\underline{0.953 \pm 0.019}$ | $\mathbf{0.961 \pm 0.005}$ |
| | Degra. Freq. (%) | $2.493 \pm 7.373$ | $83.978 \pm 10.186$ | $-$ | $\mathbf{1.049 \pm 1.545}$ | $\underline{1.825 \pm 2.732}$ |
| Unlock | AER | $0.940 \pm 0.025$ | $0.042 \pm 0.026$ | $0.258 \pm 0.157$ | $\underline{0.961 \pm 0.008}$ | $\mathbf{0.966 \pm 0.008}$ |
| | LER | $0.941 \pm 0.029$ | $0.552 \pm 0.255$ | $0.492 \pm 0.239$ | $\underline{0.968 \pm 0.008}$ | $\mathbf{0.969 \pm 0.007}$ |
| | Degra. Freq. (%) | $4.885 \pm 4.890$ | $86.588 \pm 10.775$ | $72.213 \pm 11.375$ | $\underline{0.629 \pm 0.533}$ | $\mathbf{0.217 \pm 0.172}$ |

**Preparations of Training Datasets.** We collect training datasets from the PPO algorithm training process (Schulman et al., 2017). For Dark Room, we train PPO in the same way as Algorithm Distillation. However, for Minigrid, the PPO policy faces instability when directly training from scratch. Thus, we use a pretrain-finetune mode, that is, we first pretrain a PPO model on many seeds, then finetune them on certain seed if needed. More details can be found in Appendix D.2.

**Baselines.** We use three AD-like ICRL methods, AD (Laskin et al.), AD-$\epsilon$ (Zisman et al.), and IDT (Huang et al.), as baselines. We implement all baselines as well as our method on a GPT-2 (Radford et al.) based backbone, ensuring comparable parameter scales and closely matched architectural hyperparameters. More details can be found in Appendix D.3.

**Metrics.** To quantify the frequency of performance degradation in our experiments, we propose a metric named **Degradation Frequency**, which is calculated as the proportion of episodes in which the episode reward decreases by at least 5% compared to the previous one. The Degradation Frequency $D_F$ is given by:

$$D_F = \frac{1}{N} \sum_{i=1}^{N} \mathbb{I}(r_i \leq 0.95 \cdot r_{i-1}), \tag{8}$$

where $N$ is the total number of episodes, $r_i$ is the reward of the $i$-th episode, and $\mathbb{I}(\cdot)$ is the indicator function. To evaluate the overall performance, we use **Average Episode Return** (AER) as our main metric. In addition, we also consider the **Last Episode Return** (LER), which is used in previous work (Tarasov et al.), as it reflects the final performance, indicating the ultimate in-context learning outcomes.

## 5.2 MAIN RESULTS

**The performance degradation phenomenon is prevalent in many scenarios. CV-ICRL proves effective in alleviating this issue, leading to significant improvements in overall performance.** Building on our initial observations from the Dark Room case study, we first establish that the performance degradation phenomenon is indeed widespread. Our experiments across 6 diverse tasks in the Minigrid environment confirm this prevalence. As presented in Table 1, the results for baseline methods show that significant performance degradation occurs frequently, indicated by a high Degradation Frequency (Degra. Freq.).

Against this backdrop, CV-ICRL proves highly effective. The results in Table 1 show that our method consistently and significantly lowers the Degradation Frequency across all tested tasks. This enhanced stability translates directly to superior overall performance. As illustrated in Figure 3, our

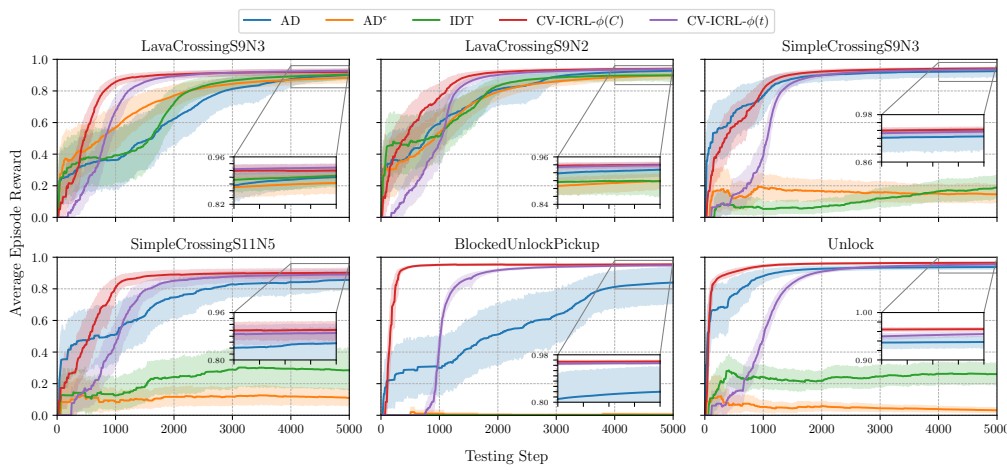

Figure 3: Testing time AER in Minigrid. The curves represent the mean AER over 20 independent tasks with the 95% confidence interval. Both of CV-ICRL-$\phi(C)$ and CV-ICRL-$\phi(t)$ demonstrate a more stable performance improvement and superior final performance.

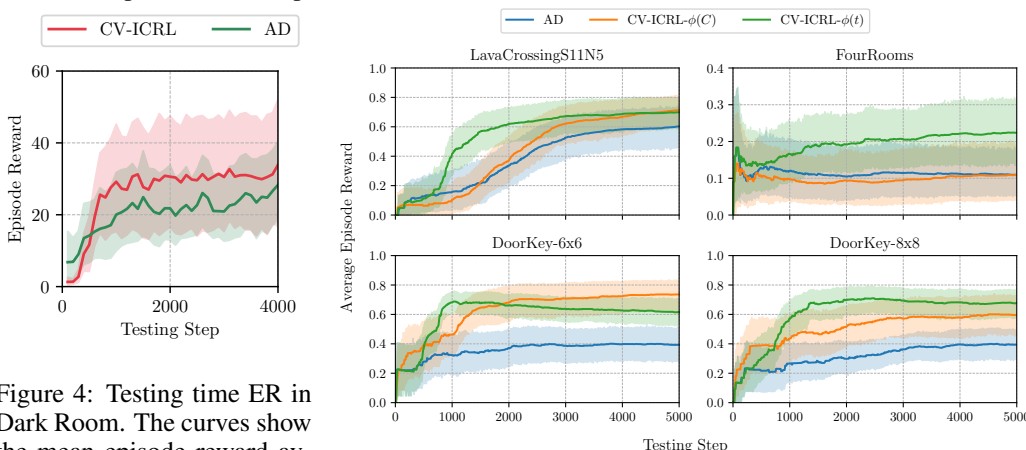

Figure 4: Testing time ER in Dark Room. The curves show the mean episode reward averaged over 20 unseen tasks. Our method (CV-ICRL-$\phi(t)$) outperforms AD in terms of episode reward.

Figure 5: Performance of ICRL policy trained on 6 task types and evaluated on 4 unseen task types in Minigrid. While AD demonstrates good generalization across varied tasks and scenes, our method improves this capability.

method's learning curves exhibit both higher final returns and smaller confidence intervals, implying more stable and reliable performance. This effectiveness is also demonstrated in the Dark Room as shown in Figure 4.

**The performance differences between the two practical CV-ICRL methods meet our expectations.** As shown in Table 1 and Figure 3, CV-ICRL-$\phi(C)$ tends to yield better overall performance, while CV-ICRL-$\phi(t)$ tends to result in a lower Degradation Frequency. This aligns with our discussion in Section 4.2, where CV-ICRL-$\phi(C)$ benefits from incorporating more context and task related information, contributing to superior performance. However, this method is also more susceptible to context ambiguity, potentially affecting stability and performance consistency. On the other hand, CV-ICRL-$\phi(t)$, maintains more monotonic behavior and results in lower degradation frequency. However, the estimate of $V_C$ in this case may deviate more significantly from the true value, leading to potential performance losses.

**The AD-like ICRL algorithm demonstrates generalization across varied tasks and scenes, and our method enhances this ability.** In prior works, ICRL methods have typically been validated in scenarios where the differences between tasks lie mainly in reward function or simple changes in the transition. Such simple modifications often result in tasks that are not significantly different from those seen in the training set, leading some work to suggest that these ICRL algorithms cannot

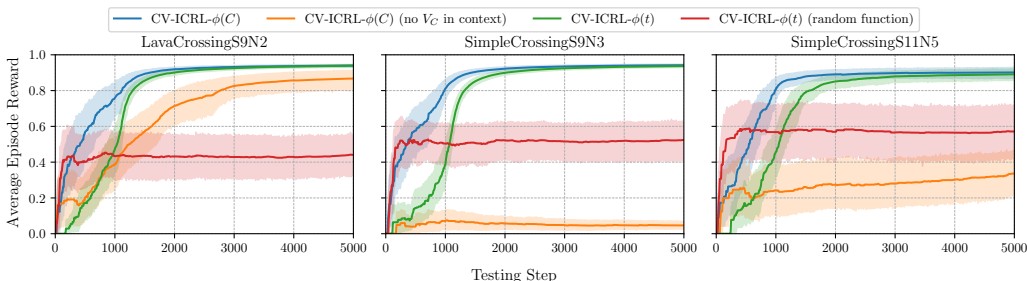

Figure 6: Ablation results. The experiments demonstrate (1) the effectiveness of integrating estimated $V_C$ into the context for CV-ICRL-$\phi(C)$, and (2) the very need of a well-designed $\phi(t)$ for CV-ICRL-$\phi(t)$.

address out-of-distribution generalization problems (Raparthy et al., 2024). In this study, we aim to validate the OOD generalization capability in the Minigrid environment. We combine data from 6 types of tasks mentioned previously into a single dataset, train a general ICRL policy, and then evaluate its performance on 4 novel types of tasks (introduced in Appendix D.1).

As shown in Figure 5, the performance on these 4 novel tasks demonstrates that although the AD-like policy has never encountered such scenarios, it still exhibits good generalization ability, effectively embodying the concept of "learn-to-learn". Moreover, our method, CV-ICRL, outperforms the AD-like policy, showing faster adaptation to novel tasks and achieving a higher average episode return.

## 5.3 ABLATIONS

We conduct ablation experiments (shown in Figure 6 and Appendix D.4) to isolate the sources of performance improvement, specifically investigating the contributions of (1) using the estimated $V_C$ for contextual guidance, and (2) the necessity of a well-designed $\phi(t)$.

**Does the performance of CV-ICRL-$\phi(C)$ improve due to the introduction of an auxiliary task or because $V_C$ is integrated into the context to guide model decisions?** To answer this question, we remove $V_C$ from the context and retrain the model, which effectively reduced the task to one involving only the auxiliary task. This version performed significantly worse than CV-ICRL-$\phi(C)$, indicating that the performance improvement is not merely due to the introduction of an auxiliary task but rather because $V_C$ is used within the context to guide decision-making.

**Does the function $\phi(t)$ in CV-ICRL-$\phi(t)$ play a crucial role given that there is no clear mapping from the context to $\phi(t)$?** To investigate this, we test the case where $\phi(t)$ is replaced with a random function. This resulted in a significant performance drop, with the AER stabilizing around a certain value. This confirms that $\phi(t)$ plays a critical role in the model's performance, and its absence or replacement with a random function severely impacts the generalization ability. We conduct further comparison experiments of $\phi(t)$ in Appendix D.4.

## 6 CONCLUSION

In this paper, we address the common performance degradation of prior ICRL methods, wherein their test-time performance fails to exhibit the monotonic improvement seen during training. We identify the root cause of this issue as *Contextual Ambiguity*, which stems from sampling randomness. To resolve this, we introduce *Context Value* as an explicit, non-decreasing signal of the ideal performance achievable given the current context. We prove that Context Value tightens the lower bound on the performance gap relative to an ideal policy. Building on this, we propose **CV-ICRL**, which incorporates practical methods for estimating this value during training and testing. Our experiments in both the Dark Room and Minigrid environments show that our method can effectively alleviate performance degradation and significantly improve the performance. Moreover, our experiments on Minigrid provide the first empirical evidence of the generalization ability of AD-like ICRL algorithms across significantly different tasks types, confirming the "learn-to-learn" capability. This insight not only contributes to the development of ICRL methods but also offers valuable

directions for future work in in-context learning in large language models (LLMs). Our approach, while grounded in traditional RL scenarios, also presents potential applications in the broader context of LLM-based in-context learning problems (Krishnamurthy et al., 2024; Tajwar et al., 2025). While CV-ICRL provides useful estimates of Context Value, a dedicated error model in future work could enhance confidence in the reported estimates. Future work can further explore more accurate estimation methods for Context Value and refine the approach to ensure stronger monotonic improvement.

**The Use of Large Language Models.** We used a large language model as a general-purpose assistant solely for text editing, including grammar correction, wording and tone adjustments, punctuation, and stylistic consistency. The model did not contribute to research ideation, methodology, experimental design, data analysis, interpretation of results, or the generation of substantive academic content or references. All suggestions were reviewed and approved by the authors, who take full responsibility for the final text.

**Ethics Statement.** Our method and algorithm do not involve any adversarial attack, and will not endanger human security. All our experiments are performed in the simulation environment, which does not involve ethical and fair issues.

**Reproducibility Statement.** The source code of this paper is available at `https://anonymous.4open.science/r/towards_monotonic_improvement-E72F`. We specify all the implementation details of our methods in Appendix D.3. The experiment additional results are in the Appendix D.4.

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

# A PROOF OF THEOREMS

## A.1 PROOF OF THEOREM 1

To formally quantify the advantage of incorporating value information, we must first establish the baseline performance. We begin by presenting the theoretical guarantee on the performance error for a standard ICRL policy, which is learned without value side-information. This result, adapted from imitation learning works (He et al.), serves as the foundation for our subsequent comparison.

**Theorem 2** (Performance Bound of ICRL Policy). The suboptimality gap for the ICRL policy, defined as the worst-case difference in expected return between the learned policy $\pi_C(\cdot|C)$ and the empirical optimal policy $\pi_C^*(\cdot|C)$, is bounded as follows:

$$\sup_C |J(\pi_C^*(\cdot|C)) - J(\pi_C(\cdot|C))| \leq \frac{2r_{\max}}{(1-\gamma)^2}\epsilon \tag{9}$$

where $\epsilon$ represents the worst-case statistical divergence between the two policies:

$$\epsilon = \sup_C D_{TV}(\pi_C^*(\cdot|C)\|\pi_C(\cdot|C)) \tag{10}$$

*Proof.* Here, we recap the definition of the expected return of $\pi_C$, i.e. expected return of $\pi^{\mathrm{ICRL}}$ for given $C$. Suppose $C_{t_0}$ is a context from $t = 0$ to $t = t_0$, then

$$J(\pi_{C_{t_0}}) = V^{\pi^{\mathrm{ICRL}}}(s_{t_0}; C_{t_0}) = \mathbb{E}_{\pi^{\mathrm{ICRL}}, T}\Big[\sum_{t=t_0}^{\infty} \gamma^t R(s_t, a_t))\Big] \tag{11}$$

Expanding $V^{\pi^{\mathrm{ICRL}}}(s; C)$ for one step:

$$V^{\pi^{\mathrm{ICRL}}}(s; C) = \sum_{a,s'} T(s'|s, a)\pi_C(a|s)[R(s, a) + \gamma V(s'; C'))] \tag{12}$$

Here, $C' = C \cup \{(a, r, s')\}$.

For notational simplicity, we use $V(s; C)$ to denote $V^{\pi^{\mathrm{ICRL}}}(s; C)$, and use $V^*(s, C)$ to denote $V^{\pi_C^*}(s; C)$ in the remainder of this proof.

Thus,

$$J(\pi_C^*) - J(\pi_C)$$

$$= \sum_{a,s'} T(s'|s, a)[\pi_C^*(a|s)[R(s, a) + \gamma V^*(s'; C'))] - \pi_C(a|s)[R(s, a) + \gamma V(s'; C'))]]$$

$$= \sum_{a,s'} T(s'|s, a)R(s, a)[\pi_C^*(a|s) - \pi_C(a|s)]$$

$$\quad + \gamma \sum_{a,s'} T(s'|s, a)[\pi_C^*(a|s)V^*(s'; C') - \pi_C(a|s)V(s'; C')]$$

$$= \sum_{a,s'} T(s'|s, a)R(s, a)[\pi_C^*(a|s) - \pi_C(a|s)]$$

$$\quad + \gamma \sum_{a,s'} T(s'|s, a)\pi_C^*(a|s)[V^*(s'; C') - V(s'; C')] + V(s'; C')[\pi_C^*(a|s) - \pi_C(a|s)]$$

$$= \sum_{a,s'} T(s'|s, a)[R(s, a) + \gamma V(s'; C')][\pi_C^*(a|s) - \pi_C(a|s)]$$

$$\quad + \gamma \sum_{a,s'} T(s'|s, a)\pi_C^*(a|s)[V^*(s'; C') - V(s'; C')]$$

$$\leq \|T(R + \gamma V)\|_\infty \|\pi_C^* - \pi_C\|_\infty + \gamma\|V^* - V\|_\infty \qquad \triangleright \text{ Hölder's inequality}$$

$$\leq 2\|(R + \gamma V)\|_\infty \epsilon + \gamma\|J^* - J\|_\infty \qquad \triangleright \text{ Equations (10) and (11)}$$

$$\leq \frac{2r_{\max}}{1 - \gamma}\epsilon + \gamma\|J^* - J\|_\infty \qquad \triangleright V(s) \leq \frac{r_{\max}}{1 - \gamma}$$

Then,

$$\|J^* - J\|_\infty \leq \frac{2r_{\max}}{1-\gamma}\epsilon + \gamma\|J^* - J\|_\infty$$

$$(1-\gamma)\|J^* - J\|_\infty \leq \frac{2r_{\max}}{1-\gamma}\epsilon$$

Then we have

$$\sup_C |J(\pi_C^*(\cdot|C)) - J(\pi_C(\cdot|C))| = \|J^* - J\|_\infty \leq \frac{2r_{\max}}{(1-\gamma)^2}\epsilon \tag{13}$$

$\square$

Here, we suppose given $C$, the estimator of optimal policy $\widehat{\pi_C^*}(a|C) = \pi^*(a|s)$ is in the training dataset. Besides, for a simpler proof and without loss of generality, we assume that the behavior policy set $\Pi$ is finite. The inference process of an ICRL model can be framed as a two-stage sampling procedure: first, sampling a policy $\pi$ from the distribution of learned source policies conditioned on the current context $C$, and second, sampling an action $a$ from the chosen policy $\pi$. This is formally expressed as a marginalization over all policies $\pi \in \Pi$:

$$\pi_C(\cdot|C) = \sum_{\pi \in \Pi} P(\pi|C, V_C)\pi(\cdot|s) \tag{14}$$

**Lemma 1.** Let the naive ICRL policy $\pi_C(\cdot|C)$ and $\pi_C(\cdot|C, V_C)$ is the ICRL policy conditioned on $V_C = J(\pi_C^*(\cdot|C))$, and $\pi_C^*(\cdot|C)$ is the optimal policy given $C$. Then

$$\sup_C D_{TV}(\pi_C^*(\cdot|C)\|\pi_C(\cdot|C, V_C)) \leq 1 - P(\pi^*|C, V_C) + k \tag{15}$$

$$\sup_C D_{TV}(\pi_C^*(\cdot|C)\|\pi_C(\cdot|C)) \leq 1 - P(\pi^*|C) + k \tag{16}$$

where $k = D_{TV}(\pi_C^*(\cdot|C)\|\pi^*(\cdot|s))$

*Proof.* Firstly, using the triangle inequality total variation distance, we have

$$D_{TV}(\pi_C^*(\cdot|C)\|\pi_C(\cdot|C, V_C)) \leq D_{TV}(\pi_C^*(\cdot|C)\|\pi^*(\cdot|s)) + D_{TV}(\pi^*(\cdot|s)\|\pi_C(\cdot|C, V_C)) \tag{17}$$

The first term is the estimation error caused by the gap between $\pi_C^*$ and $\pi^*$. Then consider the second term.

$$D_{TV}(\pi^*(\cdot|s)\|\pi_C(\cdot|C, V_C))$$

$$= \frac{1}{2}\sum_a |(\pi^*(a|s) - \pi_C(a|C, V_C)|$$

$$= \frac{1}{2}\sum_a |(\pi^*(a|s) - \sum_{\pi \in \Pi} P(\pi|C, V_C)\pi(\cdot|s)|$$

$$= \frac{1}{2}\sum_a |(\pi^*(a|s)[1 - P(\pi^*|C, V_C)] - \sum_{\pi \in \Pi\backslash\{\pi^*\}} P(\pi|C, V_C)\pi(a|s)|$$

$$\leq \frac{1}{2}\sum_a \left(|\pi^*(a|s)[1 - P(\pi^*|C, V_C)]| + \left|\sum_{\pi \in \Pi\backslash\{\pi^*\}} P(\pi|C, V_C)\pi(a|s)\right|\right)$$

$$= \frac{1}{2}\sum_a \left(\pi^*(a|s)[1 - P(\pi^*|C, V_C)] + \sum_{\pi \in \Pi\backslash\{\pi^*\}} P(\pi|C, V_C)\pi(a|s)\right)$$

$$= \frac{1}{2}\left([1 - P(\pi^*|C, V_C)]\sum_a \pi^*(a|s) + \sum_{\pi \in \Pi\backslash\{\pi^*\}} P(\pi|C, V_C)\sum_a \pi(a|s)\right)$$

$$= \frac{1}{2} \left( [1 - P(\pi^*|C, V_C)] \cdot 1 + \sum_{\pi \in \Pi \setminus \{\pi^*\}} P(\pi|C, V_C) \cdot 1 \right)$$

$$= \frac{1}{2} ([1 - P(\pi^*|C, V_C)] + [1 - P(\pi^*|C, V_C)]) \qquad \triangleright P(\pi^*|C, V_C) + \sum_{\pi \in \Pi \setminus \{\pi^*\}} P(\pi|C, V_C) = 1$$

$$= 1 - P(\pi^*|C, V_C)$$

Similarly, for the naive ICRL policy $\pi_C(\cdot|C)$ we can also have

$$\sup_C D_{TV}(\pi_C^*(\cdot|C)\|\pi_C(\cdot|C)) \leq 1 - P(\pi^*|C) + k \tag{18}$$

$\square$

Now we consider the posterior term $P(\pi|C)$. We will prove that $P(\pi^*|C, V_C)$ is larger than $P(\pi^*|C)$, then the upper bound of $D_{TV}(\pi_C^*(\cdot|C)\|\pi_C(\cdot|C, V_C))$ is tighter than the upper bound of $D_{TV}(\pi_C^*(\cdot|C)\|\pi_C(\cdot|C, V_C))$.

**Lemma 2** (Comparison of two upper bounds). Let $\pi^*$ be the estimated optimal policy of $\pi_C^*$ in training dataset, we have

$$\frac{P(\pi^*|C, V_C)}{P(\pi^*|C)} \geq 1 + \delta_{\text{rel}} \tag{19}$$

where

$$\delta_{\text{rel}} = \frac{\left( \sum_{i \neq *} P(C|\pi_i) \right) \left( e^{\beta(d_J - 2d^*)} - 1 \right)}{P(C|\pi^*)e^{\beta(d_J - 2d^*)} + \sum_{i \neq *} P(C|\pi_i)} > 0 \tag{20}$$

if $d_J - 2d^* > 0$, which means that we have a enough good estimation of $\pi_C^*$. The clear definitions of $d_J$ and $d^*$ are given in the proof.

*Proof.* Using the Bayes Rule, we have

$$P(\pi|C) = \frac{P(C|\pi)P(\pi)}{\sum_i P(C|\pi_i)P(\pi_i)} \tag{21}$$

For the assumption of training dataset, each policy $\pi$ has same prior $P(\pi) = \delta$. Thus $P(\pi|C) \propto P(C|\pi)$, where $P(C|\pi)$ is the likelihood of dataset.

Similarly, $\pi_C(\cdot|C, V_C)$ also

$$P(\pi|C, V_C) = \frac{P(C, V_C|\pi)P(\pi)}{\sum_i P(C, V_C|\pi_i)P(\pi_i)} \tag{22}$$

$$= \frac{P(C|\pi, V_C)P(V_C|\pi)}{\sum_i P(C|\pi_i, V_C)P(V_C|\pi_i)} \tag{23}$$

For a given $\pi^*$, the generation process of $C$ is independent of $V_C$, thus $P(C|\pi^*, V_C) = P(C|\pi^*)$. Then from Equations (21) and (22), we have

$$\frac{P(\pi^*|C, V_C)}{P(\pi^*|C)} = \frac{\frac{P(C|\pi^*)P(V_C|\pi^*)}{\sum_i P(C|\pi_i)P(V_C|\pi_i)}}{\frac{P(C|\pi^*)}{\sum_j P(C|\pi_j)}} = \frac{P(V_C|\pi^*)\sum_j P(C|\pi_j)}{\sum_i P(C|\pi_i)P(V_C|\pi_i)} \tag{24}$$

Here, we consider the relationship between $V_C$ and $\pi_C^*$. We model $P(V_C|\pi^*)$ as a function of the difference in values. The true value of a policy $\pi$ is $J(\pi)$. The observed empirical value is $V_C = J(\pi_C^*)$. The closer these two values are, the more likely it is that $\pi$ is the source of this observation. A common and effective model takes the form of exponential decay, similar to a Laplace or Boltzmann distribution:

$$P(V_C|\pi) \propto \exp\left(-\beta|V_C - J(\pi)|\right) = \exp\left(-\beta|J(\pi_C^*) - J(\pi)|\right) \tag{25}$$

Before analysis $P(V_C|\pi)$, we label the difference between the true value of the estimated optimal policy $\pi_C^*$ and the true optimal policy $\pi^*$ as

$$d^* = |J(\pi_C^*) - J(\pi^*)| \tag{26}$$

And the gap between the true optimal policy and the best sub-optimal policy.

$$d_J = \min_{i \neq *}\{J(\pi^*) - J(\pi_i)\} \tag{27}$$

A better training set will result in a smaller $d^*$, while a larger $d_J$ means there's a obvious gap between the estimated optimal policy and the sub-optimal one.

Then we define a ratio $\Gamma_i$ to compare the value-based likelihoods:

$$\Gamma_i = \frac{P(V_C|\pi^*)}{P(V_C|\pi_i)} = \frac{\exp(-\beta|J(\pi_C^*) - J(\pi^*)|)}{\exp(-\beta|J(\pi_C^*) - J(\pi_i)|)} = \exp\left(\beta\left(|J(\pi_C^*) - J(\pi_i)| - d^*\right)\right) \tag{28}$$

$$P(\widehat{V_C}|\pi^*) = \exp(-\beta|\widehat{V_C} - J(\pi^*)|) \tag{29}$$

Using the triangle inequality, $|J(\pi_C^*) - J(\pi_i)| \geq |J(\pi^*) - J(\pi_i)| - |J(\pi_C^*) - J(\pi^*)| \geq d_J - d^*$. Thus, we can establish a uniform lower bound $\Gamma_{\min}$:

$$\Gamma_i \geq \exp\left(\beta(d_J - 2d^*)\right) \triangleq \Gamma_{\min} \tag{30}$$

This assumes $d_J > 2d^*$, which implies $\Gamma_{\min} > 1$.

Now, we can bound the posterior ratio:

$$\frac{P(\pi^*|C, V_C)}{P(\pi^*|C)} = \frac{P(V_C|\pi^*)\sum_j P(C|\pi_j)}{P(C|\pi^*)P(V_C|\pi^*) + \sum_{i \neq *} P(C|\pi_i)\frac{P(V_C|\pi^*)}{\Gamma_i}} \geq \frac{\sum_j P(C|\pi_j)}{P(C|\pi^*) + \sum_{i \neq *} \frac{P(C|\pi_i)}{\Gamma_{\min}}} \tag{31}$$

For convenience, we label the terms related to training dataset (likelihood) $S_* = P(C|\pi^*)$ and $S_{\text{other}} = \sum_{i \neq *} P(C|\pi_i)$, then we have

$$\frac{P(\pi^*|C, V_C)}{P(\pi^*|C)} \geq \frac{S_* + S_{\text{other}}}{S_* + S_{\text{other}}/\Gamma_{\min}} \tag{32}$$

Obviously, this ratio is larger than 1, we define the relative improvement, $\delta_{\text{rel}}$ as

$$\delta_{\text{rel}} = \frac{S_* + S_{\text{other}}}{S_* + S_{\text{other}}/\Gamma_{\min}} - 1 = \frac{S_{\text{other}}(1 - 1/\Gamma_{\min})}{S_* + S_{\text{other}}/\Gamma_{\min}} = \frac{S_{\text{other}}(\Gamma_{\min} - 1)}{S_*\Gamma_{\min} + S_{\text{other}}} \tag{33}$$

Thus we have

$$\frac{P(\pi^*|C, V_C)}{P(\pi^*|C)} \geq 1 + \delta_{\text{rel}} \tag{34}$$

$\square$

**Final proof of Theorem 1.**

*Proof.* Let

$$\epsilon_{\text{base}} = \sup_C D_{TV}(\pi_C^*(\cdot|C)\|\pi_C(\cdot|C))$$

$$\epsilon_V = \sup_C D_{TV}(\pi_C^*(\cdot|C)\|\pi_C(\cdot|C, V_C))$$

We have

$$\sup_C |J(\pi^C) - J(\pi_C(\cdot|C))|$$

$$\leq \frac{2r_{\max}}{(1-\gamma)^2} \epsilon_{\text{base}} = \frac{2r_{\max}}{(1-\gamma)^2} \sup_C D_{TV}(\pi_C^*(\cdot|C)\|\pi_C(\cdot|C))$$

$$\leq \frac{2r_{\max}}{(1-\gamma)^2}(1 - P(\pi^*|C) + k) = D_{\text{base}}$$

Simialrly,

$$\sup_C |J(\pi^C) - J(\pi_C(\cdot|C, V_C))| \leq \frac{2r_{\max}}{(1-\gamma)^2}(1 - P(\pi^*|C, V_C) + k) = D_V$$

For Lemma 2, we have $P(\pi^*|C, V_C) > P(\pi^*|C)$, thus $1 - P(\pi^*|C, V_C) < 1 - P(\pi^*|C)$, thus $D_V < D_{\text{base}}$. $\qquad\square$

### A.2 PROOF OF COROLLARY 1

*Proof.* For the proofs of Theorem 2 and Lemma 1 don't use the property of $V_C$ (i.e. $V_C = \pi_C^*$), we only need to replace all the $V_C$ by $\widehat{V_C}$ in these conclusions. Now we consider the Lemma 2. Here, there is an estimation error between $\widehat{V_C}$ and $V_C$.

$$d_V = |V_C - \widehat{V_C}| = |J(\pi_C^*) - \widehat{V_C}| \tag{35}$$

Then we have

$$|\widehat{V_C} - J(\pi_i)| = |(J(\pi_C^*) - J(\pi_i)) - (J(\pi_C^*) - \widehat{V_C})|$$
$$\geq |J(\pi_C^*) - J(\pi_i)| - |J(\pi_C^*) - \widehat{V_C}|$$
$$= |J(\pi_C^*) - J(\pi_i)| - d_V$$

$$|\widehat{V_C} - J(\pi^*)| = |(\widehat{V_C} - J(\pi_C^*)) + (J(\pi_C^*) - J(\pi^*))|$$
$$\leq |\widehat{V_C} - J(\pi_C^*)| + |J(\pi_C^*) - J(\pi^*)|$$
$$= d_V + |J(\pi_C^*) - J(\pi^*)|$$

Then the ratio of likelihoods becomes

$$\Gamma_i = \frac{P(\widehat{V_C}|\pi^*)}{P(\widehat{V_C}|\pi_i)} \geq \exp\left(\beta\left((|J(\pi_C^*) - J(\pi_i)| - d_V) - (d_V + |J(\pi_C^*) - J(\pi^*)|)\right)\right)$$

$$= \exp\left(\beta\left(|J(\pi_C^*) - J(\pi_i)| - |J(\pi_C^*) - J(\pi^*)| - 2d_V\right)\right)$$

And for the conclusion in Lemma 2, we finally have

$$\Gamma_i \geq \exp\left(\beta(d_J - 2d^* - 2d_V)\right) \triangleq \Gamma_{\min} \tag{36}$$

Thus the conclusion of Theorem 1 still holds if the estimation error holds that

$$d_J - 2d^* - 2d_V > 0 \tag{37}$$

$\qquad\square$

## B   A FURTHER UNDERSTANDING OF CONTEXTUAL AMBIGUITY

Why does Contextual Ambiguity lead to a vicious cycle of performance degradation? The root cause is that the ICRL training process does not require the model to analyze context for decision-making. Instead, it encourages pattern matching. This tendency is amplified because short-term information is often more decisive for immediate actions than long-term history. As a result, the model places more weight on recent interactions. This recency bias explains why a few poor samples in the short-term context can mislead the model, making it believe it is at an earlier stage, triggering performance collapse. This is further supported by our Transformer attention heatmaps.

Due to the long horizon (i.e., 400 time steps), the attention weights are relatively small, making the visualization less pronounced. However, despite this, we can observe that for the output at time $t$, inputs close to $t$ have higher attention weights (brighter colors), suggesting that short-term information is more decisive for decision-making than long-term history.

This observation highlights why Contextual Ambiguity has such a significant impact: recent low-reward trajectories, particularly those sampled poorly, have a stronger influence on the model's decisions. It offers valuable insights into why context ambiguity leads to performance degradation and suggests potential directions for mitigating this effect in future work.

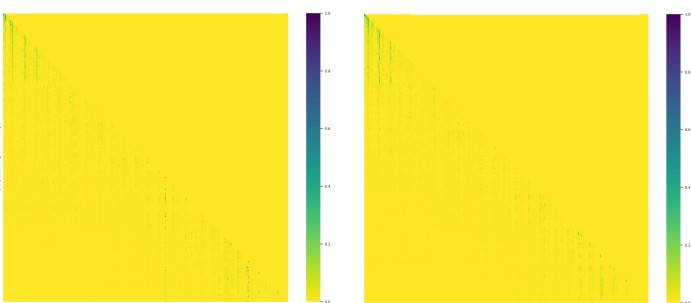

Figure 7: This heatmap represents the attention weights in the final layer of the Transformer model. Each position in the output is influenced by different positions in the input, with the heatmap visualizing the attention weights that indicate the degree of influence. Specifically, for each time step $t$ in the output sequence, the heatmap shows how much each position in the input sequence contributes to the output at that time step.

## C   Practical Algorithms

---

**Algorithm 1:** Collecting training dataset.

---

**Input:** Policies $\{\pi_1, \pi_2, \ldots, \pi_N\}_\tau$ from online RL algorithm training process for each task $\tau$.

**Output:** Training dataset $\mathcal{D}$.

1  $\mathcal{D} \leftarrow \emptyset$

2  **for** $\tau \in \mathcal{T}_{train}$ **do**

3      **for** $i$ *from 1 to N* **do**

4          Evaluating each policy $\pi_{i,\tau}$ and get $J(\pi_{i,\tau})$

5      **end**

6      **while** not reach the max number **do**

7          **for** $i$ from $1$ to $N$ **do**

8              Sampling in-context trajectories $h$ of length $H$,
               $$h_i^{(\tau)} \leftarrow (s_0, a_0, r_0, \ldots, s_{H-1}, a_{H-1}, r_{H-1}, s_H, a_H, r_H).$$

9              Adding $\widehat{V_C} = J(\pi_{i,\tau})$ at each timestep,
               $$h_i^{(\tau)} \leftarrow (s_0, \widehat{V_C}, a_0, r_0, \ldots, s_{H-1}, \widehat{V_C}, a_{H-1}, r_{H-1}, s_H, \widehat{V_C}, a_H, r_H)$$

10          **end**

11          $h^{(\tau)} \leftarrow \{h_1^{(\tau)}, h_2^{(\tau)}, \ldots, h_H^{(\tau)}\}$

12          $\mathcal{D} \leftarrow \mathcal{D} \cup h^{(\tau)}$

13      **end**

14  **end**

---

---

**Algorithm 2:** CV-ICRL-$\phi(C)$.

---

**Input:** Training dataset $\mathcal{D}$.
**Output:** Trained ICRL policy $\pi^{\text{ICRL}}$.

1 Initialize the parameters of $\pi^{\text{ICRL}}$ with $\theta$ and Context Value estimate model $\phi(V)$ with $\theta_V$.
2 **while** not converged **do**
3     Randomly sample context
       $C = (s_0, \widehat{V_C}, a_0, r_0, \ldots, s_{H-1}, \widehat{V_C}, a_{H-1}, r_{H-1}, s_H, \widehat{V_C}, a_H, r_H)$.
4     Compute the context $C_t$ for each timestep $t$ in the trajectory.
5     Update $\pi^{\text{ICRL}}$ with policy gradient based on the error between predicted and actual returns.
6     **Update rule:** Use cross-entropy loss to minimize the difference between the predicted
       return and the true return from $\pi^{\text{ICRL}}$ policy.
7     Update $\phi(V)$ (Context Value model) with Mean Squared Error (MSE) loss, based on the
       predicted Context Value $\widehat{V_C}$.
8     **Update rule:** Minimize the MSE between $\phi(C)$ and $\widehat{V_C}$.
9 **end**
10 **Testing time:**
11 Reset environment and get init state $s_0$.
12 Initialize context $C \leftarrow (s_0)$.
13 **for** $t = 0 \ldots T$ **do**
14       Compute $\widehat{V_C} = \phi(C)$ using the trained Context Value model.
15       Predict the action $a_t$ using the trained policy $\pi^{\text{ICRL}}(C)$.
16       Execute $a_t$ in the environment and observe the next state $s_{t+1}$ and reward $r_t$.
17       Update the context $C$ with $(\widehat{V_C}, a_t, r_t, s_{t+1})$.
18 **end**

---

**Algorithm 3:** CV-ICRL-$\phi(t)$

---

**Input:** Training dataset $\mathcal{D}$. Context value estimator $\phi(t)$
**Output:** Trained ICRL policy $\pi^{\text{ICRL}}$.

1 Initialize the parameters of $\pi^{\text{ICRL}}$ with $\theta$.
2 **while** not converged **do**
3     Randomly sample context
       $C = (s_0, \widehat{V_C}, a_0, r_0, \ldots, s_{H-1}, \widehat{V_C}, a_{H-1}, r_{H-1}, s_H, \widehat{V_C}, a_H, r_H)$.
4     Compute the context $C_t$ for each timestep $t$ in the trajectory.
5     Update $\pi^{\text{ICRL}}$ with policy gradient based on the error between predicted and actual returns.
6     **Update rule:** Use cross-entropy loss to minimize the difference between the predicted
       return and the true return from $\pi^{\text{ICRL}}$ policy.
7 **end**
8 **Testing time:**
9 Reset environment and get init state $s_0$.
10 Initialize context $C \leftarrow (s_0)$.
11 **for** $t = 0 \ldots T$ **do**
12       Compute $\widehat{V_C} = \phi(t)$ using the given Context Value estimator.
13       Predict the action $a_t$ using the trained policy $\pi^{\text{ICRL}}(C)$.
14       Execute $a_t$ in the environment and observe the next state $s_{t+1}$ and reward $r_t$.
15       Update the context $C$ with $(\widehat{V_C}, a_t, r_t, s_{t+1})$.
16 **end**

---

## D    DETAILS OF EXPERIMENTS

### D.1    ENVIROMNETS

#### D.1.1    DARK ROOM

The Dark Room environment is a challenging testbed for an agent's ability to perform efficient exploration under conditions of extreme reward sparsity and partial observability. The task places an agent in a large gridworld with a single goal state, but the agent's perception is limited to its local vicinity. A positive reward is only granted upon reaching the goal, meaning the agent must conduct a systematic, memory-based search to explore the space without any guiding signals. It is therefore highly effective at evaluating an agent's capacity to use memory for long-term navigation and exploration.

#### D.1.2    MINIGRID

The environments commonly employed in current methods, such as Dark Room and MiniWorld, are relatively simple. For instance, the state space in Dark Room is merely two-dimensional. More importantly, the task variations in these settings are largely confined to changes in the reward function or minor shifts in dynamics. This is insufficient to demonstrate the generalization capability of an ICRL algorithm to unseen tasks. To address this, we use Minigrid, a more complex and diverse environment, to demonstrate that our ICRL algorithm can indeed generalize to completely new types of tasks.

Minigrid environment features a collection of gridworld scenarios where an agent must infer and accomplish a goal through exploration. Tasks range from simple navigation, such as bypassing obstacles to reach a goal, to complex sequential decision-making, like opening a series of doors. Critically, Minigrid provides a fixed-size (7x7), uniform symbolic observation across all its diverse tasks. This standardization removes the challenge of unifying observation representations, making it an ideal platform to directly test how effectively an In-Context Reinforcement Learning (ICRL) policy can adapt to new tasks.

Belows are the introductions of 6 task types used for both training and testing.

**LavaCrossingS9N3.**    In this task type, the agent must navigate through a room with deadly lava streams running horizontally or vertically. The agent must reach the green goal square while avoiding lava, with a single safe crossing point for each stream. A path to the goal is guaranteed.

**LavaCrossingS9N2.**    Similar to LavaCrossingS9N3, but the challenge is slightly less difficult compared to LavaCrossingS9N3.

**SimpleCrossingS9N3.**    In this task type, the agent must navigate through a room with walls instead of lava. The objective is to reach the green goal square while avoiding walls. This task is easier compared to the LavaCrossing tasks and is useful for quickly testing algorithms.

**SimpleCrossingS11N5.**    Similar to SimpleCrossingS9N3, the agent must reach the green goal square while avoiding walls. The task involves a larger environment with more walls, making it moderately more challenging than SimpleCrossingS9N3 but still easier than the LavaCrossing tasks.

**BlockedUnlockPickup.**    In this task type, the agent must pick up an object placed in another room, behind a locked door. The door is blocked by a ball that the agent must first move to unlock the door. The agent must learn to move the ball, pick up the key, open the door, and then pick up the object in the other room.

**Unlock.**    This task type is a simplified version of BlockedUnlockPickup. The agent just need to pickup the kay and then unlock the door.

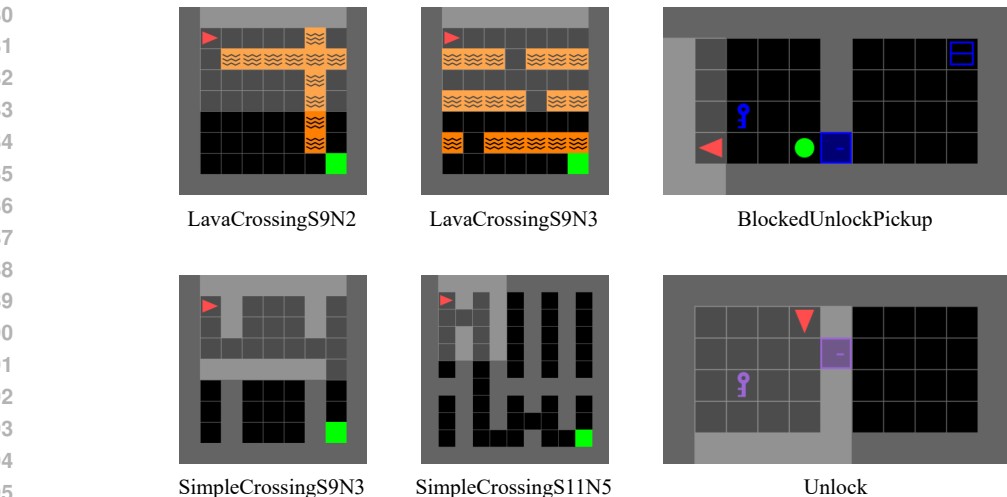

Figure 8: The 6 tasks for the main experiments, with different seeds corresponding to different layouts.

Belows are the introductions of 4 task types used for the cross-task-type generalization experiments, and only for testing.

**LavaCrossingS11N5.** Similar to LavaCrossingS9N3, LavaCrossingS11N5 introduces greater difficulty by increasing the number of lava streams and the complexity of the room layout. This makes it more challenging for the agent to find a safe path to the goal.

**FourRooms.** Agent must navigate a maze composed of four rooms, interconnected by gaps in the walls. The agent's goal is to reach the green goal square to receive a reward. Both the agent and the goal square are randomly placed in any of the four rooms.

**DoorKey-6x6.** In this task type, the agent must pick up a key to unlock a door and then reach the green goal square.

**DoorKey-8x8.** Similar to DoorKey-6x6, but is more complex due to the larger environment size.

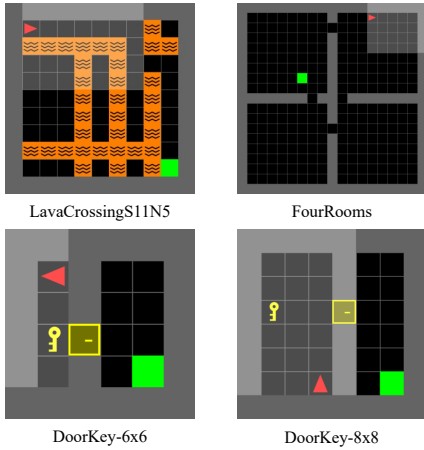

Figure 9: The 4 tasks for the cross-task generalization experiments, with different seeds corresponding to different layouts.

## D.2 PREPARATIONS OF TRAINING DATASETS

For the Dark Room environment, we train PPO in the same way as Algorithm Distillation. We choose 20 different environments, each with different goal positions (x, y), and use these to train the PPO agent.

For Minigrid, the environment is more challenging than Dark Room. The PPO policy faces instability when trained from scratch for each layout. To address this, we adopt a pretrain-finetune strategy: we first pretrain a PPO model on a variety of layouts and then finetune it on specific layouts as needed.

The PPO algorithm we use is based on the implementation provided by Stable-Baselines3, a popular library for reinforcement learning algorithms. This implementation ensures stability and efficiency during training. Below is a table summarizing the hyperparameters used for training our PPO model in both environments. The values not mentioned are set to the default values from Stable-Baselines3.

Table 2: Hyperparameters of PPO training for Dark Room

| Hyperparameter | Value |
|---|---|
| Learning Rate | 0.0001 |
| Policy Network Architecture | MlpPolicy |
| Training Steps | 1000000 |
| Save Frequency | 50000 |
| N Steps | 500 |

Table 3: Hyperparameters of PPO training for Minigrid Pre-train

| Hyperparameter | Value |
|---|---|
| Learning Rate | 0.0002 |
| Number of Epochs | 20 |
| Policy Network Architecture | CustomCNN |
| Training Steps | 2e7 |
| Save Frequency | 64000 |
| N Steps | 1600 |

Table 4: Hyperparameters of PPO training for Minigrid Fine-tune

| Hyperparameter | Value |
|---|---|
| Learning Rate | 2e-5 |
| Policy Network Architecture | CustomCNN |
| Training Steps | 400000 |
| Save Frequency | 16000 |
| N Steps | 1600 |

CustomCNN is a customized 2-layer convolutional neural network (CNN) feature extractor designed to process image inputs.

For the BlockedUnlockPickup task, due to its extreme difficulty, the model was unable to learn effectively from scratch. To address this, we used a pre-trained model from the UnlockPickup task and continued pretraining it on the BlockedUnlockPickup task.

Now, regarding the data collection process, for both Dark Room and Minigrid, we selected a series of models and arranged them to reflect their training progression: the earlier models are those that showed continuous performance improvement, while the later ones are those that had reached convergence. This arrangement was made with the intention that the ICRL policy learned from these models would also achieve stability after performance convergence. For each PPO process, we selected 40 models to sample data from a horizon of 400. For Dark Room, this process resulted in a total of 40,000 trajectories. For each task in Minigrid, we collected a total of 170,000 trajectories.

### D.3 Details of Implementations

**AD (Algorithm Distillation):** AD (Laskin et al.) is trained on continuously improving trajectories generated from the agent's experience. In this framework, the model learns to predict the next action based on a history of states, actions, and rewards, effectively transforming reinforcement learning into a supervised learning problem.

**AD$^\epsilon$:** AD$^\epsilon$ (Zisman et al.) builds upon AD by replacing real online trajectories with simulated trajectories. These simulated trajectories are generated by sampling from a noised model, with the noise progressively reduced during training. This approach allows for greater flexibility in training the model by not relying strictly on real data. It also makes the model more robust to imperfections in the trajectory data.

**IDT (In-Context Decision Transformer):** IDT (Huang et al.) extends the AD framework by re-ordering the context according to episode rewards and introducing a hierarchical decision-making structure. This hierarchical approach enables the model to handle longer horizons and more complex tasks. Specifically, IDT organizes the contexts into different decision levels, with the high-level model focusing on broader task goals, while the decision model focuses on more immediate decisions. This method allows for better handling of tasks with long temporal dependencies.

We implement all baselines as well as our method on a GPT-2 (Radford et al.) based backbone, ensuring comparable parameter scales and closely matched architectural hyperparameters.

Due to the context length limitation of our GPT-2 model, we set the context length to 400, while GPT-2 has a maximum horizon of 1024. This constraint leads to a situation where each position in the Transformer corresponds to a time step's context, rather than having continuous three or four positions in the Minigrid environment (with a 7x7x3 observation) corresponding to one timestep's context. For this, we use 2 convolutional layers followed by 1 MLP layer, embedding the observations into 64-dimensional vectors. Actions are one-hot encoded into a 7-dimensional vector, while rewards and Context Values are represented as single-dimensional values.

For AD$^\epsilon$, the model sequence length is 40. For each position $i$ in the model sequence, we set $\epsilon$ as a function: $\epsilon_i = \min\left(\frac{i}{30}, 1\right)$, which ensures consistency with AD and other algorithms in the Minigrid setting. The first 30 models correspond to trajectories with continuously improving performance, while the remaining 10 represent near-stable (optimal) models.

For IDT, we implement the hierarchical structure as outlined in the original paper. The structure includes three models: a decision model, a high-level decision model, and a reviewing decisions model. The high-level decision model's context timestep interval is set to 5, and both the decision model and high-level decision model share the same architecture and hyperparameters. The reviewing decisions model is a 2-layer MLP. The embedding size for the high-level decision model is set to 64.

For CV-ICRL, in Dark Room, we estimate the Context Value using the normalized average episode reward of the source policy. We consider the maximum average episode reward (AER) from the model sequence as 1. In Minigrid, we use the average episode reward of the source policy as an estimate for $V_C$, as the max AER for these tasks is set to 1. To evaluate the source policy's average episode reward, we average the results from 5 seeds for each source policy.

For CV-ICRL-$\phi(C)$, we add an output head parallel to the action prediction head, predicting $V_C$ with a size of 1.

For CV-ICRL-$\phi(t)$, in Dark Room, we use the function $V_C = \min\left(\frac{t}{1200}, 1\right)$. For Dark Room, we tested three different selection strategies and ultimately chose this function. For the BlockedUnlockPickup, LavaCrossingS9N2, SimpleCrossingS9N3, SimpleCrossingS11N5, and Unlock tasks, we used $\min\left(\frac{t}{1000}, 0.95\right)$ for the Context Value function. For LavaCrossingS9N3 and the cross-task generalization experiment, we used $\min\left(\frac{t}{800}, 0.875\right)$. We performed additional experiments to compare the performance of these three estimated Context Value functions.

### D.4 MORE EXPERIMENTAL RESULTS

More ablation results on another 3 Minigrid tasks.

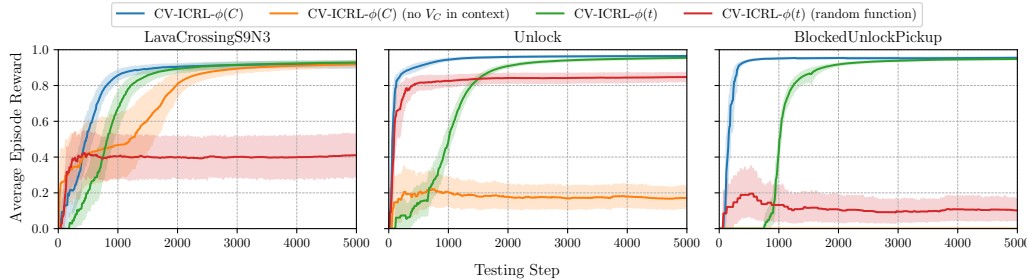

Figure 10: Additional ablation results.

Additional experimental result to compare the performance of these three estimated Context Value functions, where $\phi_1 = \min\left(\frac{t}{800}, 0.875\right)$, $\phi_2 = \min\left(\frac{t}{600}, 0.9\right)$, $\phi_3 = \min\left(\frac{t}{1000}, 0.95\right)$

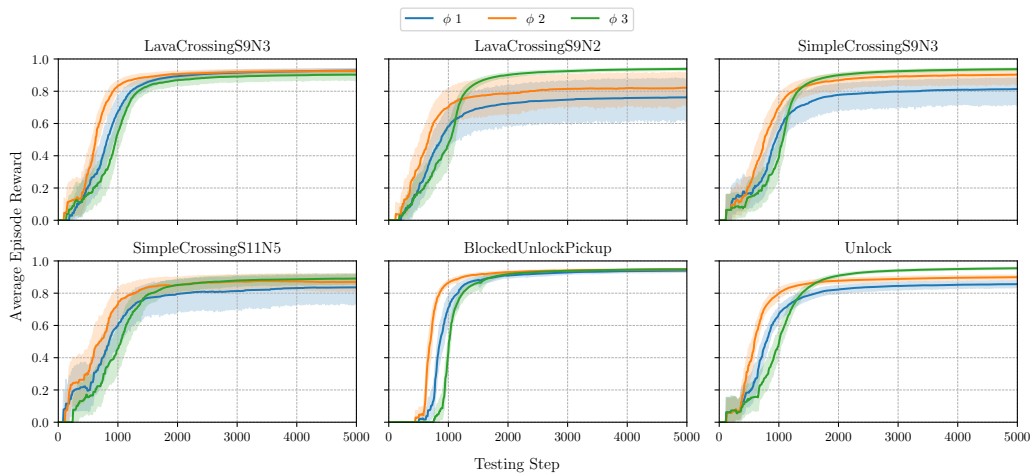

Figure 11: Comparison of $\phi(t)$.

Table 5: Addtional results for the experiments on 4 unseen task types. The best results are **in bold** and the second-best are underlined.

| Task Type | Metric | AD | **CV-ICRL-$\phi(t)$** | **CV-ICRL-$\phi(C)$** |
|---|---|---|---|---|
| LavaCrossingS11N5 | AER | $0.614 \pm 0.329$ | $0.693 \pm 0.263$ | **$0.765 \pm 0.205$** |
| | LER | $0.860 \pm 0.156$ | $0.907 \pm 0.106$ | **$0.919 \pm 0.064$** |
| | Degra. Freq. (%) | $37.364 \pm 33.672$ | $32.500 \pm 28.544$ | **$20.703 \pm 21.393$** |
| FourRooms | AER | $0.106 \pm 0.141$ | $0.219 \pm 0.200$ | **$0.277 \pm 0.231$** |
| | LER | $0.315 \pm 0.262$ | **$0.422 \pm 0.225$** | $0.288 \pm 0.320$ |
| | Degra. Freq. (%) | $85.716 \pm 11.331$ | $77.076 \pm 17.016$ | **$76.923 \pm 18.351$** |
| DoorKey-6x6 | AER | $0.464 \pm 0.222$ | $0.589 \pm 0.236$ | **$0.745 \pm 0.202$** |
| | LER | $0.629 \pm 0.275$ | $0.749 \pm 0.245$ | **$0.797 \pm 0.278$** |
| | Degra. Freq. (%) | $52.659 \pm 17.003$ | $48.098 \pm 21.714$ | **$27.007 \pm 19.836$** |
| DoorKey-8x8 | AER | $0.408 \pm 0.194$ | **$0.660 \pm 0.221$** | $0.618 \pm 0.306$ |
| | LER | $0.579 \pm 0.239$ | $0.806 \pm 0.198$ | **$0.816 \pm 0.189$** |
| | Degra. Freq. (%) | $58.258 \pm 13.199$ | **$38.563 \pm 17.920$** | $40.659 \pm 28.430$ |

