# OpenReview forum: "Towards Monotonic Improvement in In-Context Reinforcement Learning"
_ICLR.cc/2026/Conference — Submitted to ICLR 2026_

### Official Review · Reviewer_B2YQ · 2025-10-20

**Soundness:** 1
**Presentation:** 2
**Contribution:** 2
**Rating:** 2
**Confidence:** 4

**Summary:**

The authors conjecture that the performance degradation encountered in algorithm-distillation-like in-context reinforcement learning algorithms is due to having suboptimal decisions in the context due to randomness. To combat it, they propose Context Value Informed ICRL (CV-ICRL) that augments the context with the latent value of the context-optimal policy. They theoretically show that the performance bound is improved when the policy has access to the context value, and empirically demonstrate the effectiveness of CV-ICRL on Dark Room and Minigrid testbeds.

**Strengths:**

- The authors propose a novel perspective on the cause of the degradation of performance when running algorithm distillation. They term it contextual ambiguity.
- The work presents a novel algorithm called CV-ICRL that augments AD, which the authors claim addresses the contextual ambiguity.
- The empirical study demonstrates improved returns of CV-ICRL and lower performance degradation frequencies.
- The statistical analysis in the empirical study is reasonably rigorous.

**Weaknesses:**

- One major concern is the lack of rigour in the theoretical claims. For instance, I found Definition 1 ambiguous. It is unclear how a context-optimal policy is defined. One interpretation could be that the context defines an empirical MDP, and the context-optimal policy is the optimal policy that solves that MDP. However, there could be more interpretations.
- Some notations are confusing and overloaded multiple times. As an example, I've seen at least $V$, $V_C$ and $V(s; C)$ used to define different kinds of values. Sometimes, it's hard to tell if it's denoting a function or a value being mapped to.
- The contextual ambiguity is merely a conjecture. The authors did not verify if it is the real culprit. If it is the real cause, one should expect the performance degradation to disappear once the contextual ambiguity is removed.
- I am not sure if the proof of Theorem 1 is correct. Particularly, it is unclear how the authors arrive at line 645 from line 644. As far as I understand, $V$ here is a mapping from the space of state and context to a scalar, while $J$ here is a mapping from the policy space to a scalar. Therefore, I don't see how $||J^* - J||{\infty}$ is an upper bound of $ ||V^* - V||{\infty}$.
- Using the data-generating policy as a proxy for the context-optimal policy seems unjustified. It also renders the method inapplicable in cases where the policy that generates the data is missing or inaccessible.
- CV-ICRL only exhibits a relatively minor performance improvement over the AD baselines. I think it's partly due to AD can already solve most of the tasks in the testbed reasonably well. It would make a stronger case if the authors could demonstrate the robustness of CV-ICRL in tasks where AD's performance degrades significantly.

Minor concerns:
- Though not drastically affecting comprehension, there are frequent grammar and spelling errors across the text. The paper would benefit from polishing the writing.
- The paper would benefit from referencing recent surveys on in-context reinforcement learning (e.g., Moeini et al., 2025) besides meta-reinforcement learning (Beck et al., 2023).

**Questions:**

The idea of CV-ICRL conditions on the premise that the model can perform accurate policy evaluation by predicting the optimal value based on the context. It is a nontrivial task because the optimal policy can be recovered from the optimal value function. Thus, I wonder why we still wish to use the model as a policy? Why don't we directly use the model to predict, say, the optimal action values, and extract the optimal policy from them?

---

> ### Author Response · Authors · 2025-11-20
>
> We appreciate the reviewer's insightful comments and suggestions, and we'd like to clarify some misunderstandings and respond to your questions in points.
>
> **1. A clearer definition of context value.**
> > One major concern is the lack of rigour in the theoretical claims. For instance, I found Definition 1 ambiguous. It is unclear how a context-optimal policy is defined. One interpretation could be that the context defines an empirical MDP, and the context-optimal policy is the optimal policy that solves that MDP. However, there could be more interpretations.
> >
>
> We thank the reviewer for raising this concern regarding the definition of the context.
>
> In Section 3.2, we provide the commonly used definition of context in in-context reinforcement learning (ICRL) works, where the context is represented as a sequence of interaction histories: $C = \{s_0, a_0, r_0, \dots, s_t, a_t, r_t\}$.
>
> We acknowledge that the original description in Definition 1 may not have been sufficiently clear.  To make the concept of "context-optimal policy" and the meaning of "only based on C" more precise, we adopted a more formal information-theoretic definition in the revised version in lines 227-231:
>
> >$$
> \pi_C^* = \arg\max_\pi \mathbb{E}_{\tau, C, R} [R(\tau, \pi(C)],\quad \text{s.t.}\quad I(\tau;\pi)\le I(\tau;C)
> $$Here, $I(\tau;C)$ denotes the mutual information between the task and the context, and the constraint, $I(\tau;\pi)\le I(\tau;C)$ ensures that the policy does not exploit any information beyond what is available in the context.
> >
>
> **2. Clarification of notations.**
>
> > Some notations are confusing and overloaded multiple times. As an example, I've seen at least $V, V_C, V(s;C)$, and used to define different kinds of values. Sometimes, it's hard to tell if it's denoting a function or a value being mapped to.
> >
>
> > I am not sure if the proof of Theorem 1 is correct. Particularly, it is unclear how the authors arrive at line 645 from line 644. As far as I understand, here is a mapping from the space of state and context to a scalar, while here is a mapping from the policy space to a scalar.
> >
>
> We apologize for the confusion caused by our notation. The symbol $V_C$ denotes the **context value** of context $C$，whereas every $V$ without a subscript refers to the **value function of a policy**. For example, $V(s;C)$ represents the value function of the ICRL policy given the context $C$. We acknowledge that, for brevity, we simplified $V^{\pi^{\text{ICRL}}}(s;C)$ to $V(s;C)$，which may have led to confusion and made it harder to distinguish from $V_C$。We will clarify this point in the proof section to avoid misunderstanding.
>
> In the proof of Theorem 1, $J^* - J$ is a shorthand for the expression $J(\pi_C^*)-J(\pi_C)$, and the definition of $J(\pi_C)$ is given in Equation 11, i.e. $J(\pi_{C_{t_0}}) = V^{\pi^{\text{ICRL}}}(s_{t_0};C_{t_0})$.  Since $\pi_C$ is a policy conditioned on $C$, $J(\pi_C)$ differs from the usual $J(\pi)$: it is a mapping from both the policy and the context (including $s_{t_0}$) to a scalar, similar to $V^\pi.$ We have clarified this point to avoid further misunderstanding in the revised version.
>
> **3. Explaination of Contextual ambiguity.**
> > The contextual ambiguity is merely a conjecture. The authors did not verify if it is the real culprit. If it is the real cause, one should expect the performance degradation to disappear once the contextual ambiguity is removed.
> >
>
> Thank you for the thoughtful comment. To clarify, "contextual ambiguity" in our paper is not a conjectured explanation, but refers to a concrete phenomenon arising at test time: because the available context is limited and stochastically sampled, the resulting context distribution inevitably deviates from the training distribution. This distribution shift can make a model trained on the training distribution fail to adapt accurately at test time. Thus, contextual ambiguity captures the mismatch between the training and test context distributions that naturally occurs under limited test-time sampling.
>
> Furthermore, contextual ambiguity cannot be removed in a straightforward manner, because contexual ambiguity does not arise solely from how contexts are constructed, but from both the inherent limitations of test-time sampling and the learned ICRL policy. Our introduction of the context value is motivated by the need to mitigate the impact of this unavoidable distribution mismatch, rather than assuming that it can be completely removed.

---

> ### Author Response · Authors · 2025-11-20
>
> **4. Clarification of our methods.**
> > Using the data-generating policy as a proxy for the context-optimal policy seems unjustified. It also renders the method inapplicable in cases where the policy that generates the data is missing or inaccessible.
> >
> We would like to clarify that the source policy used in our method is not exactly the context-optimal policy. Instead, it serves as an estimation of the context-optimal policy. This follows the standard practice in prior ICRL work (e.g., AD [1], IDT [2]), where the policy obtained from the online RL (PPO) process is treated as an approximate reference. In our setting, both the source policy and the contexts are derived from the same training data distribution; an ideal online RL policy would extract near-optimal behavior from this distribution, making it a natural estimator for the context-optimal policy.
>
> We would like to clarify that the absence of the data-generating policy does not render our method inapplicable. Even when the policy is missing or inaccessible, $J(\pi)$ can still be estimated from data, although such estimates may carry higher variance or bias. This consideration partly motivated the development of CV-ICRL-$\phi(t)$, which aims to reduce the dependence on $J(\pi)$.
>
> [1] Laskin, Michael, et al. "In-context Reinforcement Learning with Algorithm Distillation." *The Eleventh International Conference on Learning Representations*.
>
> [2] Huang, Sili, et al. "In-Context Decision Transformer: Reinforcement Learning via Hierarchical Chain-of-Thought." *Forty-first International Conference on Machine Learning*.
>
> **5. Emphasizing the performance improvement.**
> > CV-ICRL only exhibits a relatively minor performance improvement over the AD baselines. I think it's partly due to AD can already solve most of the tasks in the testbed reasonably well. It would make a stronger case if the authors could demonstrate the robustness of CV-ICRL in tasks where AD's performance degrades significantly.
> >
>
> While the improvements in Average Episode Reward on LavaCrossingS9N3, LavaCrossingS9N2, SimpleCrossingS9N3, and Unlock are relatively modest, the results across the remaining six environments in Figure 3 and Figure 5 (including the OOD settings) show **clear and substantial gains**. These gains indicate that CV-ICRL is able to successfully solve tasks that AD and other baselines consistently fail on, demonstrating a meaningful advantage in both capability and robustness.
>
> Moreover, we would like to emphasize the **large reduction** in **Degradation Frequency** reported in Table 1. The Degradation Frequency is computed as the proportion of episodes in which the episode reward decreases by at least $5\%$ compared to the previous one, as defined in Eq.(7)(line 349). This metric directly captures the performance degradation phenomenon that our method aims to address, and the improvements are both consistent and significant. The results highlight that CV-ICRL not only enhances the achievable performance but also **substantially improves the stability and reliability** of ICRL policies.
>
> **6. Grammar and Spelling Corrections.**
> > Though not drastically affecting comprehension, there are frequent grammar and spelling errors across the text. The paper would benefit from polishing the writing.
> >
>
> We apologize for the grammar and spelling errors. We have carefully proofread the paper and corrected grammatical and spelling issues we found. We also revised several sentences to enhance clarity, readability, and overall writing quality.
>
> **7. Reference update.**
> > The paper would benefit from referencing recent surveys on in-context reinforcement learning (e.g., Moeini et al., 2025) besides meta-reinforcement learning (Beck et al., 2023).
> >
>
> We thank the reviewer for the helpful suggestion. We are familiar with Moeini et al. (2025), *"A Survey of In-Context Reinforcement Learning,"*[3] and we appreciate its comprehensive overview of the field. We have added a reference in the revised version to provide a broader perspective on related survey works.
>
> [3] Moeini, Amir, et al. "A survey of in-context reinforcement learning." arXiv preprint arXiv:2502.07978 (2025).

---

> ### Author Response · Authors · 2025-11-20
>
> **8. Additional clarification of context value.**
> > The idea of CV-ICRL conditions on the premise that the model can perform accurate policy evaluation by predicting the optimal value based on the context. It is a nontrivial task because the optimal policy can be recovered from the optimal value function. Thus, I wonder why we still wish to use the model as a policy? Why don't we directly use the model to predict, say, the optimal action values, and extract the optimal policy from them?
> >
>
> Thank you for the comment. We would like to offer a clarification of context value. It is important to distinguish between **context value** and the **value of a policy.** The latter represents the expected return of taking action $a$ in state $s$ and is directly used for decision-making. In contrast, the context value we define is **policy-agnostic**—it does not encode the optimal policy nor can it directly guide action selection. Context value is to quantify the information contained in the context, not to evaluate or determine a control policy. We hope this clarification helps align the interpretation of context value with the intended definition in our work.

---

### Official Review · Reviewer_1G4v · 2025-10-29

**Soundness:** 1
**Presentation:** 1
**Contribution:** 1
**Rating:** 0
**Confidence:** 4

**Summary:**

The work proposes a modification to in context reinforcement learning via a so called context value. This context value is introduced to avoid ambiguities of contexts, by providing a clear label for individual context windows. The proposed context value based ICRL is evaluated for algorithm distillation style ICRL methods. In the empirical evaluation the work shows that their proposed method outperforms the algorithm distillation baseline. The idea of using a context value to avoid disambiguities of contexts is promising and could potentially inform further meta-RL research. However, I believe that the work is far from publication in it's current form. I am doubtful of some of the theoretical ideas as well as the empirical evaluation. Overall I vote for rejection.

**Strengths:**

The idea of using a context value to avoid disambiguities of contexts is promising and could potentially inform further meta-RL research.

**Weaknesses:**

I believe that a more thorough discussion of related work would be needed as meta-RL and ICRL methods are not just limited to the AD style approaches. Take for example the work by Melo (https://proceedings.mlr.press/v162/melo22a.html) which modifies the RL$^2$ paradigm to work with transformers. A follow-up on this work showed that cross-episode attention (through a hierarchical transformer architecture) essentially avoids disambiguities in context and enables better learning (https://openreview.net/forum?id=UENQuayzr1). Besides, contextual RL provides a different notion on what context means. In this setting, context is used to learn general policies/value functions to be able to adapt in a zero-shot setting. Works like "Contextualize Me" (https://openreview.net/forum?id=Y42xVBQusn) e.g. show how context-optimal policies behave and how "Context Values" can be used to learn general policies.

When discussing the contextual ambiguity, I fail to see why it is reasonable to assume monotonic improvement, especially when training across different tasks. Policies that are specialized to solve one task perfectly are much more likely to fail on another dissimilar task. Thus the monotonicity assumption seems to likely be false.

Further, Property 1 says that the context value is monotonic simply since adding another tuple to the context provides more information. This seems extremely wrong to me. The additional tuple could provide redundant data which would not increase the information about the task or irrelevant information or otherwise not add anything. In some cases adding more data can actually decrease information content. So I fail to see why the monotonicity should hold for the context value.

When discussing CV-ICRL with estimation of Vc through the context it is stated the reward-to-go is used as supervision signal. Since this value is then used as part the context to the agent. How exactly is that different to the reward-to-go used as context in the decision transformer?

The generalization experiments do not test for out-of-distribution generalization. Out of the 4 "novel" test tasks only the four rooms environment comes close to testing out of distribution capabilities as it is possible to observe states without any walls. All other environments are in the training distribution. Thus the experiments are testing interpolation capabilities but definitely not out-of-distribution capabilities. The survey by Kirk et al (https://www.jair.org/index.php/jair/article/view/14174/26890) provides a clear evaluation protocol for RL to assess out of distribution capabilities. I believe the claim (of the conclusion) that AD-like ICRL algorithms are capable of "learning-to-learn" does not hold.

I fail to see the utility of the ablation studies. While I appreciate that not including $\phi(C)$ provides insights about performance gains from auxiliary task training, not using any other target besides the reward to go would be more informative. Similarly, choosing a random function can not provide any meaningful insights at all. Since $\phi(t)$ was proposed to focus on the monoticity assumption, choosing *any other monotonic function* would be more informative than a random function.

**Questions:**

* How exactly is that different to the reward-to-go used as context in the decision transformer?
* Why were only AD style methods considered?
* Why did you not consider a combination of $\phi(C)$ and $\phi(t)$ that tries to incorporate both task knowledge as well as monotonicity?

---

> ### Author Response · Authors · 2025-11-20
>
> We appreciate the reviewer's insightful comments and suggestions, and we'd like to clarify some misunderstandings and respond to your questions in points.
>
> **1. Why only considered AD style approaches?**
>
> > I believe that a more thorough discussion of related work would be needed as meta-RL and ICRL methods are not just limited to the AD style approaches. Take for example the work by Melo (https://proceedings.mlr.press/v162/melo22a.html) which modifies the RL paradigm to work with transformers. A follow-up on this work showed that cross-episode attention (through a hierarchical transformer architecture) essentially avoids disambiguities in context and enables better learning (https://openreview.net/forum?id=UENQuayzr1). Besides, contextual RL provides a different notion on what context means. In this setting, context is used to learn general policies/value functions to be able to adapt in a zero-shot setting. Works like "Contextualize Me" (https://openreview.net/forum?id=Y42xVBQusn) e.g. show how context-optimal policies behave and how "Context Values" can be used to learn general policies.
> >
>
> > Why were only AD style methods considered?
> >
>
> Thank you for pointing out these related approaches. Our work specifically focuses on improving AD style ICRL methods, one of the two major lines of ICRL research: supervised pretraining (offline) and RL-based pretraining (online) [6]. In the related work section, we also discuss several ICRL approaches that differ substantially from AD.
>
> We are aware of the Meta-RL methods mentioned by the reviewer (e.g., Melo[1] and ECET[2]). While these methods operate under settings that resemble ICRL, they are built upon MAML- or R²-style meta-learning frameworks and are fundamentally **online** meta-RL algorithms. This forms a different line of research from the AD-style methods we target, which follow an **offline supervised-pretraining** framework and do not require online environment interactions when training transformers. For this reason, we see them as conceptually related but not directly comparable within the scope of this work, and we include only AD-like offline supervised-pretraining methods as baselines, as they share the same training paradigm and provide the most direct and fair point of comparison for evaluating our contributions.
>
> Regarding the notion of *context*, it is natural that different research communities adopt different definitions. In Contextualize Me[3], the context provides extra signals beyond the observation to describe the MDP. In contrast, our work follows the standard definition in the in-context RL literature, where context refers to the interaction history between the agent and the environment (as formally defined in Section 3.2). We hope this clarifies that our definition is consistent with the conventions of the ICRL setting we study.
>
> [1] Melo, Luckeciano C. "Transformers are Meta-Reinforcement Learners." *International Conference on Machine Learning*. PMLR, 2022.
>
> [2] Shala, Gresa, et al. "Efficient Cross-Episode Meta-RL." *The Thirteenth International Conference on Learning Representations*. 2025
>
> [3] Benjamins, Carolin, et al. "Contextualize Me–The Case for Context in Reinforcement Learning." *Transactions on Machine Learning Research*.
>
> **2. Reason for assuming monotonic improvement.**
>
> > When discussing the contextual ambiguity, I fail to see why it is reasonable to assume monotonic improvement, especially when training across different tasks. Policies that are specialized to solve one task perfectly are much more likely to fail on another dissimilar task. Thus the monotonicity assumption seems to likely be false.
> >
>
> We would like to clarify that the notion of monotonic improvement in our analysis refers exclusively to within-task improvement as more context is provided on the same task. This assumption is simply that, more informative context should not make an agent perform worse on the same task, which we provide a detailed explanation in our response to the next question (3.The monotonicity of context value).
>
> This has no connection to whether a policy trained on one task will succeed or fail on a different, dissimilar task. Our argument of monotonicity is not about transferring skills across tasks, but about the expected improvement when adapting within a single task.
>
> This assumption is also implicit in many ICRL methods, such as AD[4] and IDT[5], which are both trained on trajectories that themselves exhibit monotonic improvement over time (i.e., expert or online-RL demonstrations that continually improve). The purpose of such training data is precisely to encourage the learned policy to adopt the "learn from history and improve" pattern.

---

> ### Author Response · Authors · 2025-11-20
>
> In addition, as shown in Figure 2, we empirically observe that even on tasks **seen during training**, methods like AD can exhibit **performance degradation** when additional context is provided. This degradation is therefore not a consequence of cross-task failure, but the difficulty of estimating how informative a given context is under testing-time stochasticity and limited sampling.
>
> [4] Laskin, Michael, et al. "In-context Reinforcement Learning with Algorithm Distillation." *The Eleventh International Conference on Learning Representations*.
>
> [5] Huang, Sili, et al. "In-Context Decision Transformer: Reinforcement Learning via Hierarchical Chain-of-Thought." *Forty-first International Conference on Machine Learning*.
>
> **3. The monotonicity of context value.**
>
> > Further, Property 1 says that the context value is monotonic simply since adding another tuple to the context provides more information. This seems extremely wrong to me. The additional tuple could provide redundant data which would not increase the information about the task or irrelevant information or otherwise not add anything. In some cases adding more data can actually decrease information content. So I fail to see why the monotonicity should hold for the context value.
> >
>
> We need to clarify that the context value in our paper is non-decreasing. As defined in Definition 1, the context-optimal policy is an idealized policy that fully utilizes all information available in the context. Under this definition, adding more context cannot reduce the usable information. Formally in Property 1, we let $C$ be the current context and $C’$ an additional context, **both sampled from task $\tau$**. Then the mutual information $I(\tau; C, C') \ge I(\tau; C)$, implying that the oracle value cannot decrease.
>
> Addtionally, we don't assume strong monotonicity. Additional context may be redundant or uninformative, so context value is not expected to improve strictly with every added piece of context.
>
>
> **4. Differences between context value and reward-to-go.**
> > When discussing CV-ICRL with estimation of Vc through the context it is stated the reward-to-go is used as supervision signal. Since this value is then used as part the context to the agent. How exactly is that different to the reward-to-go used as context in the decision transformer?
> >
>
> > How exactly is that different to the reward-to-go used as context in the decision transformer?
> >
>
> Reward-to-go (or return-to-go) is computed only from the rewards already obtained in the context and the target reward of the task, and therefore it cannot represent "the value of the context itself". It is fundamentally different from the context value. For example, even if the context already contains substantial information about the environment but has not yet reached the goal (and thus has not received the terminal reward), the RTG remains 0, while the context value has already increased.
>
> **5. Clarification of OOD experiments.**
> > The generalization experiments do not test for out-of-distribution generalization. Out of the 4 "novel" test tasks only the four rooms environment comes close to testing out of distribution capabilities as it is possible to observe states without any walls. All other environments are in the training distribution. Thus the experiments are testing interpolation capabilities but definitely not out-of-distribution capabilities. The survey by Kirk et al (https://www.jair.org/index.php/jair/article/view/14174/26890) provides a clear evaluation protocol for RL to assess out of distribution capabilities. I believe the claim (of the conclusion) that AD-like ICRL algorithms are capable of "learning-to-learn" does not hold.
> >
> We need to clarify that these experiments are indeed out-of-distribution, as the evaluation tasks introduce the transitions and observation structures that are absent from the training distribution. Among the four novel tasks we use, *Doorkey* requires the agent to pick up the key to unlock the door and then reach the goal, and this type of task is not included in the training distribution. In *LavaCrossingS11N5*, the agent observes more numerous and more complex lava strips, and it can also encounter situations where no walls are visible in the observation.

---

> ### Author Response · Authors · 2025-11-20
>
> **6. Clarification of ablation studies.**
> > I fail to see the utility of the ablation studies. While I appreciate that not including $\phi(C)$ provides insights about performance gains from auxiliary task training, not using any other target besides the reward to go would be more informative. Similarly, choosing a random function can not provide any meaningful insights at all. Since $\phi(t)$ was proposed to focus on the monoticity assumption, choosing *any other monotonic function* would be more informative than a random function.
> >
>
> >Why did you not consider a combination of $\phi(C)$ and $\phi(t)$ that tries to incorporate both task knowledge as well as monotonicity.
> >
>
> In our ablation study, the purpose of using a random function is to show that introducing a monotonic function  $\phi(t)$, which is independent of the context, is indeed effective. Moreover, as you mentioned, we also experimented with different monotonic functions $\phi(t)$, and the corresponding results are provided in Figure 11 in Appendix D.4.
>
> Our goal in proposing these two methods, $\phi(C)$ and $\phi(t)$, is to provide possible estimation strategies, and we have demonstrated their effectiveness. While certain combinations or hybrids of these two strategies might yield additional gains, such combinations are not central to our method and fall outside the primary focus of this work. As noted in the conclusion, exploring improved or more sophisticated estimation strategies remains an interesting direction for future research.

---

> > ### Comment · Reviewer_1G4v · 2025-11-20
> >
> > Thank you for the detailed responses to all reviewers.
> > I just wanted to acknowledge that I have read all other reviews and responses and remain unconvinced. I will keep my score as is.

---

### Official Review · Reviewer_nRch · 2025-11-01

**Soundness:** 2
**Presentation:** 4
**Contribution:** 3
**Rating:** 4
**Confidence:** 3

**Summary:**

This paper proposes Context Value Informed ICRL (CV-ICRL), introducing a context value signal during training, which explicitly represents the ideal, theoretically achievable performance given the current context. It is argued that algorithm distillation-like ICRL algorithms work best with monotonically improving context, similar to their training data, while actual test-time contexts often contain noisy behaviors. The authors give theoretical results showing the addition of CV yields a tighter bound for performance, and empirically validates CV-ICRL in grid-world environments, especially regarding task generalization.

**Strengths:**

- Addresses an important problem in ICRL. Robustness to noise is critical under long-context and OOD scenarios.

- Provides theoretical guarantees about the benefits of the introduced context value.

- Empirical results show good overall performance with OOD task generalizations.

**Weaknesses:**

- The definition of context value is a bit vague. How to compute the optimal policy "only based on C, without any other information of $\tau$"?

- CV-ICRL seems to use the source policy for each episode of context in its training data as the context-optimal policy. There should be some explanations as to why this property holds.

- Experiments are conducted in very simple grid-world environments.

**Questions:**

- How to operationalize Def.1? Is there a way to practically compute the context-optimal policy?

- Arguments about the correctness of training-time context values are missing, and seem to require some non-trivial assumptions (e.g. dataset contains only optimal policies for each environment)

---

> ### Author Response · Authors · 2025-11-20
>
> We appreciate the reviewer's insightful comments and suggestions, and we're pleased to hear that you value our work. We'd like to clarify some misunderstandings and respond to your questions in points.
>
> **1. A clearer definition of context value.**
> > The definition of context value is a bit vague. How to compute the optimal policy "only based on C, without any other information of "?
> >
>
> > How to operationalize Def.1? Is there a way to practically compute the context-optimal policy?
> >
>
> Thank you for pointing this out. We acknowledge that the original description in Definition 1 may not have been sufficiently clear. To make the concept of "context-optimal policy" and the meaning of "only based on C" more precise, we adopted a more formal information-theoretic definition in the revised version in lines 227-231:
>
> >$$
> \pi_C^* = \arg\max_\pi \mathbb{E}_{\tau, C, R} [R(\tau, \pi(C)],\quad \text{s.t.}\quad I(\tau;\pi)\le I(\tau;C),
> $$where $I(\tau;C)$ denotes the mutual information between the task and the context, and the constraint, $I(\tau;\pi)\le I(\tau;C)$ ensures that the policy does not exploit any information beyond what is available in the context.
> >
>
> Because such a policy generally cannot be computed directly from the context alone, our method focuses instead on estimating the context value, which is a more tractable quantity. The two estimation strategies described in Section 4.2 CV-ICRL-$\phi(C)$ and and CV-ICRL-$\phi(t)$ provide practical approximations of this context value and allow the learner to infer policies aligned with the information content of the context. We hope this clarification helps make Definition 1 and the role of context value more precise.
>
> **2. Source policy is the estimation of context-optimal policy.**
>
> > CV-ICRL seems to use the source policy for each episode of context in its training data as the context-optimal policy. There should be some explanations as to why this property holds.
> >
>
> > Arguments about the correctness of training-time context values are missing, and seem to require some non-trivial assumptions (e.g. dataset contains only optimal policies for each environment)
> >
>
> We would like to clarify that the source policy used in our method is **not** exactly the context-optimal policy. Instead, it serves as an estimation of the context-optimal policy. This follows the standard practice in prior ICRL work (e.g., AD [1], IDT [2]), where the policy obtained from the online RL (PPO) process is treated as an approximate reference. In our setting, both the source policy and the contexts are derived from the same training data distribution; an ideal online RL policy would extract near-optimal behavior from this distribution, making it a natural estimator for the context-optimal policy.
>
> Such estimation inevitably introduces estimation errors. Accordingly, in the proof of **Corollary 1** (Appendix A.2 — note the numbering error), we denote the estimation errors for the context-optimal policy and the context value as $d^*$ and $d_V$, respectively. We further show that when these estimation errors are reasonably small, our improvement performance bound still holds.
>
> **3. Experiments.**
> > Experiments are conducted in very simple grid-world environments.
> >
>
> Thank you for raising this concern. We would like to clarify that these environments are far from trivial. Due to partial observability and highly sparse rewards, standard online RL algorithms (e.g., PPO) fail to learn the optimal policy, as shown in Appendix D.2.
>
> Moreover, compared to commonly used grid-world environments in prior ICRL work (e.g., Darkroom or Miniworld used by AD[1], IDT[2], etc.), our tasks involve substantially more challenging exploration dynamics, with more complex action spaces, observation spaces, and task objectives. We believe this makes them a meaningful step forward in evaluating ICRL methods.
>
>
>
> [1] Laskin, Michael, et al. "In-context Reinforcement Learning with Algorithm Distillation." *The Eleventh International Conference on Learning Representations*.
>
> [2] Huang, Sili, et al. "In-Context Decision Transformer: Reinforcement Learning via Hierarchical Chain-of-Thought." *Forty-first International Conference on Machine Learning*.

---

### Official Review · Reviewer_8tXY · 2025-11-02

**Soundness:** 3
**Presentation:** 2
**Contribution:** 3
**Rating:** 6
**Confidence:** 3

**Summary:**

This paper deals with in-context reinforcement learning (ICRL) and proposes Context Value Informed ICRL to reduce Contextual Ambiguity. They show that context value tightens the lower bound on the performance gap when considering idealistic monotonically improving policy. They perform experiments across two benchmarks - Dark Room and Minigrid - to demonstrate that their approach works better.

**Strengths:**

### Strengths:

1. This work identifies and introduces “Contextual Ambiguity” which denotes that even random action taken at early interaction can generate an interaction history that may mislead to assume different context.

2. They prove an improved performance bound with the guidance from the context value.

3. Ablations are conducted legitimately, and the experimental results are encouraging to show the efficacy of the proposed approach.

**Weaknesses:**

### Weaknesses:

1. Without enough history, the context value in test time still could be sub-optimal. Do you have any bound on how much history it would need to clearly identify the context?

2. While the paper validates the approach on two benchmarks, it is limited to very grid like setup. It would require more thorough experiments across other types of tasks to demonstrate wide applicability. Also, apart from AD-like baselines, comparisons with other strong baselines would strengthen the work.

3. Further, the memory or computational overhead due to these additional components has not been discussed. Also, I would like to see some discussion on the limitations.

**Questions:**

1. Is there any new hyperparameter that is introduced to learn the context-value?

---

> ### Author Response · Authors · 2025-11-20
>
> We appreciate the reviewer's insightful comments and suggestions, and we're pleased to hear that you value our work. We'd like to clarify some misunderstandings and respond to your questions in points.
>
> **1. Sub-optimality of context value.**
>
> > Without enough history, the context value in test time still could be sub-optimal. Do you have any bound on how much history it would need to clearly identify the context?
> >
>
> Thank you for the thoughtful question. We would like to clarify that, the sub-optimality in estimating the context value at test time does not stem from having an insufficient amount of history. Increasing the history length does not necessarily make the estimate more accurate. Instead, the gap arises from the limitations of the test-time estimation procedures themselves. As discussed in Section 4.2, both estimation strategies may introduce approximation errors.
>
> **2. Experiments on more envs.**
>
> > While the paper validates the approach on two benchmarks, it is limited to very grid like setup. It would require more thorough experiments across other types of tasks to demonstrate wide applicability. Also, apart from AD-like baselines, comparisons with other strong baselines would strengthen the work.
> >
>
> Thank you for this valuable suggestion. Our chosen environments, however, already represent a meaningful step beyond the commonly used settings in prior ICRL work (e.g., AD[1], IDT[2] on Darkroom or Miniworld), offering substantially higher structural complexity and more challenging exploration dynamics.
>
> That said, broader empirical validation on additional task families or settings could further enhance the scope of our evaluation. As potential candidates, environments built on the MuJoCo physics simulator or Procgen[3] (which offers more complex and diverse tasks) appear promising directions for extending the experimental coverage. We would also be very interested in learning which specific task domains the reviewer considers most representative or informative for assessing ICRL methods. This feedback is valuable, and we will take it into consideration when planning future experimental extensions.
>
> [1] Laskin, Michael, et al. "In-context Reinforcement Learning with Algorithm Distillation." *The Eleventh International Conference on Learning Representations*.
>
> [2] Huang, Sili, et al. "In-Context Decision Transformer: Reinforcement Learning via Hierarchical Chain-of-Thought." *Forty-first International Conference on Machine Learning*.
>
> [3] Cobbe, Karl, et al. "Leveraging procedural generation to benchmark reinforcement learning." Proceedings of the 37th International Conference on Machine Learning. 2020.
>
>
> **3. More details about our method.**
>
> > Further, the memory or computational overhead due to these additional components has not been discussed. Also, I would like to see some discussion on the limitations.
> >
>
> > Is there any new hyperparameter that is introduced to learn the context-value?
> >
>
> In terms of network architecture, predicting the context value only adds an additional output head to the original causal Transformer, which introduces no significant memory or computational overhead. We also do not introduce any new tunable hyperparameters, as the added value loss is incorporated without requiring additional loss weights or training coefficients.
>
> Regarding limitations, although our method has demonstrated effectiveness across multiple environments, scaling it to richer or more realistic real-world environments may require additional engineering and domain-specific adjustments.

---

### Author Response · Authors · 2025-12-03

Dear AC,

We thank the AC and all reviewers for their careful reading. Our paper pinpoints a central failure mode of current In-Context RL methods: despite being trained on data with monotonic policy improvement, sequence models can suffer from performance degradation at test time. We show this degradation is not accidental but systematic, arising from a contextual ambiguity phenomenon. To resolve this issue, we introduce context value (CV) as a policy-agnostic signal that measures the task-relevant informativeness of a context, and formalize context-optimality to characterize the ideal performance achievable under a given context. Building on this formulation, we provide a theoretical analysis showing that CV yields a tighter lower bound on the performance gap to an ideal monotonically improving policy. On the algorithmic side, we propose Context Value-Informed ICRL (CV-ICRL), which incorporates CV into training as an explicit target, ensuring that longer contexts are used in a non-degrading, performance-consistent way. Empirically, across multiple environments and unseen test settings, CV-ICRL delivers substantial gains in performance and robustness, most notably by sharply reducing the frequency of performance degradation and improving the stability of test-time improvement.

Across the reviews, the reviewers’ questions and concerns fall into three themes. First, the definition of context value was viewed as insufficiently clear and not concrete enough, and several reviewers were unconvinced by the stated (non-decreasing) monotonicity property, asking for a more precise explanation of what the quantity measures and under what conditions monotonicity should hold. Second, reviewers questioned our estimation of both context value and the context-optimal policy, especially the justification for using source/data-generating policies as proxies and the reliability or potential bias of the proposed estimation procedures. Third, reviewers felt that the empirical gains were not always compelling and that the chosen environments did not fully demonstrate OOD generalization, noting that improvements over AD-style baselines appeared modest in some cases or that the evaluation settings might reflect interpolation rather than clear distributional shift.

In our revised version and responses, we made targeted changes to address each concern. We provide fully formalized definitions of context value and the context-optimal policy, stressing that the latter is an idealized oracle introduced for analysis to avoid misunderstanding about what “only based on context” means. We also clarify why the source policy serves as a practical estimator (not an oracle) of the context-optimal policy, and discuss when this proxy is appropriate and what estimation bias/variance it may induce. We cleaned up notation overload, added step-by-step explanations in the proof, and made the role of estimation error explicit for readability and rigor. On the empirical side, we clarify why the four novel tasks constitute OOD shift relative to training, because they introduce transition dynamics and observation structures not seen in the training set. We also reiterate that our environments go beyond prior ICRL testbeds in structure and exploration difficulty. Finally, we emphasize that the key empirical takeaway is not only mean-reward improvement but also the large and consistent reduction in degradation frequency, which directly targets the failure mode we study.


With these clarifications and revisions, we consider the reviewers’ technical questions fully addressed, and negative assessments rooted in setting misunderstandings no longer apply under the clarified formulation. We respectfully ask the AC to evaluate the remaining critiques in light of the corrected problem framing and revised definitions.

Overall, our work identifies contextual ambiguity as a systematic cause of test-time degradation in ICRL and addresses it with a principled notion of context value. By formalizing context-optimality and using CV as a policy-agnostic training target, we obtain a tighter theoretical characterization of the gap to ideal monotonic improvement and propose an algorithm (CV-ICRL) that enforces non-degrading use of longer contexts. The revised paper and responses clarify the definition, monotonicity scope, and estimation assumptions, and demonstrate benefits on both in-distribution and genuinely OOD tasks, most notably through consistent reductions in degradation frequency alongside performance gains. Together, these conceptual, theoretical, and empirical contributions represent a meaningful advance for In-Context Reinforcement Learning, and we sincerely hope the paper will be considered favorably for acceptance.

Sincerely, Authors.

---

### Meta-Review · Area_Chair_p64t · 2025-12-12

**Summary:**

This paper proposes a new method to address the contextual ambiguity problem for in-context RL. I have the following major concerns.

(1) I think this submission lacks theoretical rigour. Many reviewers point out that the very first concept "context-optimal policy" is not well defined in the initial version. Even in the revised version, I still don't think it's well defined. It seems in response to the reviews, the authors introduce the mutual information between task and context in this definition. But this is never precisely defined. Furthermore, this mutual information appears only in this definition and never appears again. Moreover, I don't really understand what "Property X" means. There is no precise proof for this property and only hand-waving argument in human language. One reviewer also has more other back-and-forth about theoretical clarity.  Overall, the lack of theoretical rigour makes it impossible to evaluate the theoretical contribution of this work.
(2) I agree with one reviewer that the problem contextual ambiguity is not well established. There are many other possible hypothesis to explain the phenomenon observed in test time, e.g., the policy could just be exploring. I suggest the authors to spend some effort to rigorously identify this problem first.
(3) This is related to (2) but I don't think A1.3 makes sense (this is also a concern of reviewer). Even we consider the most successful RL algorithm, there is no monotonic improvement guarantee in practice, especially when exploration is considered.
(4) Given all the weakness above, the fate of this submission largely depends on the real empirical improvements. Unfortunately, experiments are only considered in tabular worlds. At least the AD paper has DeepMind Lab. And out-of-distribution case is not sufficiently evaluated (I acknowledge it's evaluated though)

I, therefore, recommend reject. (I acknowledge that 1G4v may have some misalignment with the authors so down-weighted their review significantly)

**Reviewer Concerns:**

see summary for outstanding reviewer concerns

**Reviewer Scores:**

I don't think any reviewer will change the score.

---

### Decision · Program_Chairs · 2026-01-26

Reject